# An Insight into the Defects-Driven Plasticity in Ductile Cast Irons

**DOI:** 10.3390/ma16103748

**Published:** 2023-05-15

**Authors:** Giuliano Angella, Marcello Taloni, Marcin Górny, Jacek Tarasiuk, Sebastian Wronski, Roberto Montanari, Matteo Pedranz, Matteo Benedetti, Vigilio Fontanari, Danilo Lusuardi

**Affiliations:** 1Institute of Condensed Matter Chemistry and Technology for Energy (CNR-ICMATE), Department of Chemical Science and Materials Technology (DSCTM), Via R. Cozzi 53, 20125 Milan, Italy; talonimarcello92@gmail.com; 2Faculty of Foundry Engineering, AGH University of Science and Technology, Władysława Reymonta 23, 30-059 Krakow, Poland; mgorny@agh.edu.pl; 3Faculty of Material Science and Ceramics, AGH University of Science and Technology, Al. Mickiewicza 30, 30-059 Krakow, Poland; tarasiuk@agh.edu.pl (J.T.); wronski@fis.agh.edu.pl (S.W.); 4Department of Industrial Engineering, University of Rome Tor Vergata, Via del Politecnico, 1, 00133 Roma, Italy; roberto.montanari@uniroma2.it; 5Doctoral Program in Industrial Innovation, Department of Industrial Engineering, University of Trento, Via Sommarive 9, 38123 Trento, Italy; matteo.pedranz@unitn.it (M.P.); matteo.benedetti@unitn.it (M.B.); vigilio.fontanari@unitn.it (V.F.); 6Fonderie Ariotti SpA, Via E. Fermi, 5, 25030 Adro, Italy; danilo.lusuardi@fonderieariotti.com

**Keywords:** ductile irons, chunky graphite, integrity of castings, tensile strain hardening, quality index

## Abstract

The microstructure and tensile behavior of two heavy section castings that had chemical compositions typical of GJS400 were investigated. Conventional metallography, fractography, and micro-Computer Tomography (μ-CT) were employed, enabling the quantification of the volume fractions of eutectic cells with degenerated Chunky Graphite (CHG), which was identified as the major defect in the castings. The Voce equation approach was exploited to evaluate the tensile behaviors of the defective castings for integrity assessment. The results demonstrated that the Defects-Driven Plasticity (DDP) phenomenon, which refers to an unexpected regular plastic behavior related to defects and metallurgical discontinuities, was consistent with the observed tensile behavior. This resulted in a linearity of Voce parameters in the Matrix Assessment Diagram (MAD), which contradicts the physical meaning of the Voce equation. The findings suggest that the defects, such as CHG, contribute to the linear distribution of Voce parameters in the MAD. Furthermore, it is reported that the linearity in the MAD of Voce parameters for a defective casting is equivalent to the existence of a pivotal point in the differential data of the tensile strain hardening data. This pivotal point was exploited to propose a new material quality index assessing the integrity of castings.

## 1. Introduction

Defects and metallurgical discontinuities form during metallic alloys solidification and can affect significantly the magnitude and the variability of mechanical properties. In castings of Ductile Irons (DIs), for instance, the main defects and discontinuities can be degenerated graphite agglomerates, dross, inclusions, gas and solidification shrinkage porosities, etc., which can have a detrimental impact on several mechanical properties, such as the room-temperature tensile strength, ductility, fatigue resistance, and fracture toughness [1,2,3,4,5,6,7]. In the continuous casting production of steels, for instance, the metallurgical discontinuities, such as cracks forming during solidification at the slab surface, influence appreciably the material quality resulting from the subsequent slab straightening [8,9]. In this case, hot tensile strength and ductility investigations have been reported to be effective tools to investigate the transformations from liquid to solid state that can foster the crack formation during solidification [10,11].

In DIs, one of the most common forms of degenerated graphitic agglomerates is chunky graphite (CHG), which consists of an interconnected coral-like structure that is significantly different from the ideal spheroidal graphitic agglomerates and hinders the crack nucleation and propagation. Conversely, CHG has been reported to be profuse on the fracture surface of tensile and fatigues specimens, as it fosters the crack nucleation and significantly accelerates the crack propagation [12,13]. CHG results from undercooling occurring at the interface between graphite and molten metal rich in silicon [14,15,16], so the risk of CHG formation increases with lower solidification rates, typically encountered in heavy sections [15,17,18]. In fact, the probability of degeneration in heavy section castings increases when the solidification time is longer than 1 h [17,18]. Even in the presence of a silicon content of 4.2 wt%. and higher, CHG has been found in 50 and 75 mm thick Y-blocks produced in compliance with ASTM A 536–84(2019)e1 [19,20]. The chemical composition, as well, plays an important role, and there is an extensive effort to properly control the addition of micro-alloying elements in order to avoid CHG formation. A high content of Si enhances the CHG formation, but also traces of Ce, Ca, Ni, and rare elements [14,21,22,23,24,25], while chemical elements such as Sb, Sn, Bi, and Pb hinder the CHG development [26,27,28]. However, it should be highlighted that despite the numerous studies on the CHG formation mechanism, a generally accepted theory on the formation of CHG still does not exist.

Indeed, it is not simple to quantify the detrimental effect of the CHG on tensile properties in comparison with other defects or metallurgical discontinuities [12,29]. In heavy sections, where cooling rates are slow, other defects, such as gas porosities, solidification shrinkage porosities, and inclusions, may also be present to different extents, depending on the molten treatment and chemical composition [14,21,22,23,24,25,26,27,28], thus making it difficult to quantify the defects and, among them, to assess which defect may be more detrimental. Furthermore, the properties of the metallic matrix can make it challenging to determine the contribution of microstructural defects to mechanical properties. For instance, in High Silicon Strengthened DIs (HSiSDIs), where CHG formation is common with increasing Si, the presence of chemical ordering [22,23] has been suggested to decrease ductility. Additionally, as the Si content increases, more chemical ordering is expected. Therefore, accurately quantifying the detrimental effects of CHG or chemical ordering on mechanical properties is not straightforward.

The analysis of the tensile strain hardening at room temperature in defective materials using the dislocation-density-related constitutive equation of Voce, which is based on physical principles, can provide valuable information regarding the defective nature of DIs [19,20,30]. The Voce constitutive equation is:(1a)σ=σV+σo−σV·exp−εPεc
where *σ* and *ε_P_* are the true stress and plastic strain, respectively; *σ_V_* is the saturation stress or maximum stress; *ε_C_* is the critical plastic strain that defines the rate with which the saturation stress is achieved; and *σ_o_* is the back-extrapolated stress to zero plastic strain, close to yield stress. Voce parameters in Equation (1a) can be found from the analysis of the tensile strain hardening data through the differential form of the constitutive Voce equation, defined as:(1b)$$\frac{d\sigma }{d\varepsilon_{p}}=\Theta_{o}-\frac{\sigma }{\varepsilon_{c}}$$

Equation (1b) has physical meaning, as 1/*ε*_c_ is the softening term related to dynamic recovery that comes into dislocation annihilation and formation of low-energy dislocation structures [31], and *Θ*_o_ is the dislocation multiplication factor due to dislocation storage at internal deformation structures and grain boundaries, and according to the Kocks–Mecking model, *Θ*_o_ is inversely related to the dislocation cells’ dimensions, grain size and other obstacles to dislocation motion.

In a plot of d*σ*/d*ε_P_* vs. *σ*, called a Kocks and Mecking plot [19,20,30,31] (here on called K-M plot), data linearity can be found at high stresses well in the plastic regime corresponding to Stage III of strain hardening, and fitted with Equation (1b), so finding the constant Voce parameters *Θ*_o_ and 1/*ε*_c_. Stage II of strain hardening is found only in single crystal experiments, while Stage II cannot be found in tensile tests, because deformation localization starts before the transition from Stage II to Stage IV [31]. An example of Voce parameters calculation is reported in Figure 1 for a typical tensile flow curve of GJS400. In Figure 1a, the procedure to find the Voce parameters in the K-M plot through the strain hardening analysis by using the differential Voce equation (Equation (1b)) to fit the linear region of Stage III of strain hardening is reported, while in Figure 1b, the corresponding tensile flow curve is shown. In the K-M plot, *Θ* is the strain hardening rate (d*σ*/d*ε_P_*), where *σ* has the usual meaning.

The Voce parameters 1/*ε*_c_ vs. *Θ*_o_ are plotted into the *Matrix Assessment Diagram* (MAD), and different DIs with different chemical compositions and production routes can be uniquely identified [19,20], while the integrity of the casting can be assessed with the *Integrity Assessment Diagram* (IAD) that is built by plotting the strain to rupture *ε_Rupture_* vs. *ε_Uniform_*, namely, the uniform strain, according to the Voce formalism:(2)$$\varepsilon_{Uniform}=\varepsilon_{c}\cdot ln\left[\left(\frac{\varepsilon_{c}+1}{\varepsilon_{c}}\right)\left(\frac{\sigma_{V}-\sigma_{o}}{\sigma_{V}}\right)\right]$$

When *ε_Rupture_* is lower than *ε_Uniform_*, premature failure occurs because of defects and metallurgical discontinuities, while for *ε_Rupture_* larger than *ε_Uniform_*, the ductile iron can be considered sound.

Indeed, when the Voce parameters 1/*ε*_c_ and *Θ*_o_ obtained from the tensile strain hardening analysis of a meaningful statistical set of tensile flow curves of DIs are plotted in MAD, the Voce parameters lie on a line both for sound and defective castings; however, in sound DIs, the intercept of the best fitting line is positive, while it is negative in defective materials. The finding that in defective materials the Voce parameters lie on a line, too, is quite surprising since a random effect on plastic behavior might have been expected, with a resulting increase of Voce data scattering rather than a regular linear trend. Indeed, the regularity of the Voce parameters’ linear trend in MADs for defective materials allowed the proposal of the neologism *Defects-Driven Plasticity* (DDP). The regular plastic behavior of defective materials seems to suggest that a diffuse distribution of defects or metallurgical discontinuities, such as CHG or porosity, for instance, should be responsible for this unexpected regular plastic behavior, according to the continuous damage mechanics, where the formation and growth of voids or cracks in the microstructure should occur during plastic deformation, forming a stable crack growth before the final rupture by instability [32,33]. Conversely, a single severe defect in a sound material could be also responsible for the final rupture by instability because of stress intensification producing a brittle fracture [33]. In this second case, the defect is expected to drive the fracture event itself rather than contributing to the whole plastic behavior. However, though the correlations between defective DIs and the intercepts of the best linear fits of Voce parameters has been established, so far, the DDP has not been rationalized yet.

This study aims to investigate the impact of CHG on the tensile behavior and integrity assessment procedure of two distinct heavy sections with the chemical compositions of GJS400, featuring a silicon content of approximately 2.5 wt%. Due to the high presence of CHG in these heavy sections, conventional metallographic and fractography techniques were unable to accurately quantify the extent of CHG. In addition to conventional techniques, micro-Computer Tomography (μCT) was employed to classify graphite aggregates, quantify the volume fractions where spheroidal nodules were present, and determine the volume fractions where CHG was dominant. The Voce approach [19,20,30] was used to analyze the tensile strain hardening behavior of castings, and the Voce equations parameters were used for the material quality assessment based on the MAD and IAD. New insights into the correlation between defective DIs and MAD are presented in this study, and the latest findings are used to propose a new quality index.

## 2. Experimental Section

### Materials and Microstructure Characterization

Two heavy sections with GJS400 grade chemical compositions were investigated; the chemical compositions of the relevant elements are reported in Table 1. The chemical analyses were carried out using an Optical Emission Spectrometer (OES) on DI tablets obtained by rapid solidification of the melts in order to have white cast irons with no graphite that could have interfered with OES measurements. Additionally, other elements, such as Cu, Ni, and Ti, were checked according to standard cast irons foundry practice, resulting in quantities < 0.01 wt%, while the O and N contents were not measured. The first DI (code GJS400_P) was produced in a single prismatic block with the geometry 600 × 400 × 200 mm^3^ and was affected by a very low cooling rate. The second GJS400 (code GJS400_Ce) had some Ce that came from adding 0.2% of Ce inoculation with the intention of degenerating the nodular graphite and fostering the formation of CHG, and was produced into a cylindrical block of 300 mm diameter and 520 mm height. The chemical composition of the third GJS400 (code GJ400_Y), produced through Y blocks with 50 mm and 75 mm thicknesses, in compliance with ASTM A 536-84(2019)e1 [34], which was investigated in [30,35,36], is also reported for comparison purposes in Table 1.

To assess the presence of CHG graphite and other defects, and to carry out fractography, observations were conducted using a Scanning Electron Microscope (SEM) SU-70 produced by Hitachi, Tokyo, Japan. Furthermore, the presence of CHG and degenerated graphite in the investigated heavy sections was found to be very high, especially in samples from the GJS400_P casting. However, as CHG metallographic observations are inherently areal and highly dependent on the cross-section of the metallographic sample, a volumetric technique was required to obtain statistically reliable data on the presence of CHG in the tensile specimens. Therefore, micro-Computer Tomography (μCT) was employed to obtain volumetric quantitative measurements related to CHG, which could then be correlated with the tensile mechanical behavior. The μCT measurements presented in this paper were performed using the Nanotom 180S device produced by GE Sensing & Inspection Technologies phoenix|X-ray Gmbh, Wunstorf, Germany. This device is equipped with a nanofocus X-ray tube with a maximum of 180kV voltage. The tomograms were registered on a Hamamatsu 2300 × 2300 pixel detector. The reconstruction of measured objects was accomplished with the aid of the proprietary GE software datosX v2.4.0, using the Feldkamp algorithm for cone beam X-ray CT [37]. All examined specimens were scanned at 170 kV of source voltage and a 200 μA tube current, with a rotation of the specimen of 360 degrees in 1800 steps. The copper 0.5 mm filter was used. The exposure time was 500 ms, and a frame averaging of 5 and image skip of 1 were applied, resulting in a scanning time of 120 min. The reconstructed images had a voxel size of (5 µm)^3^. The post reconstruction data treatment was performed using VG Studio Max 3.1 [38] and free Fiji-win64 software [39].

With μCT measurements, it was possible to discriminate between metallic matrix and graphite agglomerates because of their different densities. However, due to the small size of CHG in eutectic cells located close to the instrumentation resolution and their high density in eutectic cells, it was difficult to quantify CHG agglomerates accurately. As a result, a different strategy was implemented. The eutectic phase full of fine CHG and the ferritic phase in the micro-tomographic image differed slightly in brightness: the former was darker, while the latter was brighter. The difficulty in identifying the phases was caused by the relatively high level of noise in the raw tomographic image. Therefore, the images were denoised using a median filter, and after denoising, a top-hat filter [40] was additionally applied, which highlighted the differences between the areas. Then, the images were thresholded to extract only the eutectic or only the ferritic part. The number of voxels in each part was then counted and multiplied by the voxel volume. The voxel volume is equal to the measurement resolution raised to the power of three. In this way, the volumes of each phase were determined; thus, the algorithm could discriminate between the two volumes with significantly different mean densities. The first volume, referred to as *V_Ferrite_*, had a higher mean density and corresponded to the ferritic matrix with large embedded nodular graphitic agglomerates. The second volume, known as *V_Eutectic_*, had a lower mean density and corresponded to the eutectic cells filled with CHG.

The DIs were tensile tested: round tensile specimens with an initial gauge length *l*_o_ = 28.0 mm and diameter *d*_o_ = 5.6 mm were machined off and tensile tested in compliance with standard ASTM E8/E8M-11 [41] at a constant strain rate of 10^−4^ s^−1^. The engineering stress *S* = *F*/*A*_o_, where *F* is the applied force and *A*_o_ is the initial gauge cross-sectional area, and elongation *e* = (*l* − *l*_o_)/*l*_o_, where *l* is the instantaneous gauge length, were transformed into true stress *σ* = *S*∙(1 + *e*) and true strain *ε* = ln(1 + *e*). Only the true plastic strain *ε_p_* was considered for the strain hardening analysis, where *ε_p_* = *ε* − *ε_el_*, with *ε_el_* = *σ*/*E* and *E* the elastic Young’s modulus.

The tensile specimens from the different castings were so taken:prismatic heavy section casting GJS400_P—two tensile specimens called C_1 and C_2 were machined off from the core of the block; two additional specimens, named S_1 and S_2, were taken from close to the external surface of the block, so materials from different cooling conditions could be tested;cylindrical heavy section casting GJS400_Ce—10tensile specimens were machined off from a 25 mm thick slice at about the 200 mm height of the cylindrical block, with codes ranging from Nr 11 to Nr 20;50 and 75 mm blocks GJS400_Y—details of the specimen selection and tensile testing are provided in [30,35,36].

## 3. Results

### 3.1. Microstructure

Figure 2 presents representative metallographic micrographs of the grounded and polished samples of GJS400_P taken with Backscattered Electrons Imaging (BEI). Figure 2a,b display selected microstructures of the heads of the C_2 and E_2 tensile specimens, respectively, from both the core and external parts of the heavy prismatic section. The microstructures in both samples were found to be similar, showing a significant presence of fine CHG and a few areas with large graphitic agglomerates embedded in the ferritic matrix. In Figure 2a,b, very large CHG eutectic cells are visible, which have been outlined with white dotted lines for clarity.

In sample C_2, the CHG cell structure appeared to be more defined, with larger cells. On the other hand, the microstructure of E_2 was finer and more confusing, with smaller and less defined eutectic cells. Figure 3a,b show typical CHG microstructure details at higher magnifications. In sample C_2 (Figure 3a), the CHG is very fine within the eutectic cells, while at the eutectic cell boundaries, coarse CHG is visible. Figure 3b shows that in sample E_2, a few graphitic agglomerates were found in the ferritic structure between the eutectic cells, providing clear evidence of graphite degeneration. The few visible graphite nodules had diameters of about 100 µm. No signs of solidification shrinkage porosity were observed on the metallographic samples. This indicates that the relevant defects present in the material were likely limited to CHG.

Figure 4 and Figure 5 present selected representative micrographs of the microstructure of samples from the GJS400_Ce casting. The presence of CHG was not dramatic, as seen in the GJS400_P casting. The shape of the graphitic agglomerates was quite irregular, indicating that the addition of Ce was effective in promoting the formation of both CHG and degenerated graphite, as shown in Figure 4 for the tensile specimen heads of samples Nr 17 and 20. A few eutectic cells were found, with typical fine CHG within them and coarse CHG at the boundaries, as shown in Figure 5 for sample Nr 16.

In contrast, the reference GJS400_Y casting exhibited an excellent graphitic microstructure, with nodularity exceeding 83% and a pearlitic areal fraction less than 3.5%. Moreover, no signs of CHG were observed in the microstructure (for more details, see [30]).

### 3.2. Tensile Properties

Tensile tests were carried out on the 4 tensile specimens from the GJS400_P casting, namely, in-house codes C_1, C_2 from the prism core, E_1 and E_2 from the external part of the prism, and 10 tensile specimens from the GJS400_Ce casting, namely, in-house codes 11 to 20. The engineering flow curves are illustrated in Figure 6a for the GJS400_P casting, and in Figure 6b, for the GJS400_Ce casting, where *S* (MPa) is the engineering stress and *e* (%) the elongation, while the relevant tensile mechanical properties of all tensile specimens, i.e., Yield Stress (YS), Ultimate Tensile Stress (UTS), and elongations to rupture, are reported in Table 2. Figure 6a shows the tensile behavior of the specimens from the core, namely, C_1 and C_2. The tensile behavior of these specimens was found to be consistent and significantly different from that of specimens E_1 and E_2. The flow curves from tensile specimens C_1 and C_2 had a mean UTS = 291.2 MPa and a mean elongation to rupture = 2.71%, while specimens E_1 and E_2 had a mean UTS = 363.8 MPa and a mean elongation to rupture = 5.42%. Conversely, the mean YS were quite similar, resulting in 261.1 MPa and 281.8 MPa for the C and E specimens, respectively. Differently, the specimens from the GJS400_Ce casting presented a wide range of tensile mechanical properties, with UTS spanning from 321.9 MPa to 461.4 MPa, and elongations to ruptures from 2.53% to 17.9%, while the mean YS was 270.8 MPa with a standard deviation of 2.2 MPa, indicating that the YS values had little scatter. The Young’s moduli seemed to be higher in the C specimens of the GJS400_P casting than in specimens from GJS400_Ce; however, because of the poor statistics, no conclusive considerations could be drawn.

Voce parameters were calculated and are reported in Table 2, namely, *Θ*_o_, 1/*ε*_c_, *σ*_o_ and *σ_V_*, and *ε_Rupture_*, which is the last plastic strain at which the Voce equation matches the experimental flow curve at high stresses, and then rupture occurred, while *ε_Uniform_* has the usual meaning. The Voce parameters *Θ*_o_ and 1/*ε*_c_ are plotted in the MADs reported in Figure 7a and Figure 8b for tensile specimens from GJS400_P and GJS400_Ce, respectively. In Figure 9a, the Voce parameters of the sound GJS400 produced in Y-blocks with 25 mm Lynchburg, and 25, 50, and 75 mm Y-blocks (data published in [30]), are reported for comparison. In Figure 7b, Figure 8b, and Figure 9b, the data *ε_Rupture_* vs. *ε_Uniform_* are plotted in IADs.

In MADs, the scales of the different castings in Figure 7a, Figure 8a, Figure 9a are very different, as in the GJS400_P casting with a high density of CHG, the Voce parameters had wide possible values. As the presence of defects was reduced by the transitioning from the GJS400_P casting to the GJS400_Ce casting and finally to the sound GJS400_Y casting, the possible values of the Voce parameters decreased significantly. The scales of the different castings in Figure 7b, Figure 8b, Figure 9b are different in the IADs, but they follow an opposite trend, in which the maximum strains were 0.08 for the highly defective GJS400_P casting, increased to 0.16 for the GJS400_Ce casting, and reached 0.24 for the sound GJS400_Y casting. In addition, the differences between the defective and sound GJS400 castings were evident. The *ε_Rupture_* increased as the integrity of the casting improved. Furthermore, the data points were consistently located below the bisector line in the defective castings, while in the sound GJS400_Y castings, they were significantly above the bisector line.

In MADs, the best fitting lines had negative intercepts for defective castings, namely, −18.6 for the GJS40_P casting, where numerous eutectic cells full of CHG were found (Figure 2 and Figure 3); −8.8 in GJS400_Ce, where a few eutectic cells and copious CHG in the ferrite matrix were found. Conversely, in the sound GJS400_Y casting, the intercept was positive, equal to +3.62.

### 3.3. Fractography

Figure 10 shows representative fracture surface micrographs of sample C_1 from the GJS400_P casting, taken through SEM with Secondary Electron Imaging (SEI). In Figure 10a, a general view of the fracture surface reveals that the fracture was predominantly fibrous, with signs of plastic deformation. Graphitic agglomerates in DIs operate as voids that later grow in dimensions with straining [42,43]; as the CHG size in the eutectic cells was very fine (see Figure 3), highly deformed cavities and small dimples were found, resulting in a noticeably fibrous fracture surface, as reported in Figure 10b, which provides details of a ductile fracture region at higher magnification. The density of CHG on the fracture surface appeared very high, consistent with the fact that the fracture path follows the graphitic agglomerates. The last fracture region, where cleavage facets were visible, indicating fracture by instability, is shown in Figure 10c, and graphite nodules can be observed. The presence of very few shrinkage porosities on the fracture surface, as seen in Figure 10d, suggests that CHG is probably the only significant defect in the material being investigated. Figure 11 shows representative fracture surface micrographs of sample C_2, also taken through SEM with SEI. In Figure 11a, a general view of the fracture surface indicates that the fracture was generally ductile, which is also visible in Figure 11b at higher magnification, where a high amount of CHG is present. The last fracture region is depicted in Figure 11c, revealing cleavage facets and graphite nodules, and in Figure 11d at higher magnification.

Figure 12 shows representative fracture surface micrographs of sample E_1 from the GJS400_P casting. Figure 12a reveals that the fracture was predominantly ductile, which is also evident in Figure 12b at higher magnification, where a high amount of CHG is present. As in the previous cases, very few shrinkage porosities were found on the fracture surface. Figure 13 illustrates representative fracture surface micrographs of sample E_2, showing a general view in Figure 13a of ductile behavior. Figure 13b, at higher magnification, depicts the presence of a high amount of CHG in the ductile fracture, and nodules in the brittle last fracture with cleavage facets. A few shrinkage porosities were observed on the fracture surface. The fracture surfaces of the core specimens (C_1 and C_2) and external specimens (E_1 and E_2) from the GJS400_P casting appeared indistinguishable, presenting a significant quantity of CHG and only a few cleavage fracture areas where nodules were visible. Therefore, although CHG played a determinant role in the plastic and fracture behavior, fractography could not suggest how much CHG affected the plastic behavior of the samples from the GJS400_P casting, which led to the distinct flow curves reported in Figure 6 and different Voce parameters in MAD and IAD.

Fracture surfaces of selected samples from the GJS400_Ce casting are presented in Figure 14, Figure 15 and Figure 16. The selection of these samples was based on the position of the Voce parameter points in the MAD shown in Figure 8a, where tensile samples Nr 17 and Nr 20 occupied the lowest and highest positions, respectively, while Nr 16 was in between. Looking at the corresponding flow curves, sample Nr 17 had the best ductility, achieving necking; sample Nr 20 had the worst ductility; and sample Nr 16 had intermediate ductility. Figure 14 shows the fracture surface of sample Nr 17, where no sign of CHG was detected, and the surface appeared to be covered in ferritic matrix and nodules. Most of the nodules appeared to be cut into two halves, suggesting that the graphitic agglomerates were degenerated because of Ce. No evidence of solidification shrinkage porosities was found on this surface.

Figure 15 illustrates the fracture surface of sample Nr 16. As shown in Figure 15a, the fracture extended into two different regions: one with a ferritic matrix and nodules (rounded with yellow line), and a second one covered in CHG. The inset reported in Figure 15b points out details of the fracture surface covered in CHG and the region with a ferritic matrix with nodules, while no evidence of solidification shrinkage porosities was found.

On the other hand, as shown in Figure 16, the fracture surface of sample Nr 20 is almost entirely covered in CHG, with only limited regions presenting a ferritic matrix with nodules. No evidence of shrinkage porosities was found. Therefore, for the samples from the GJS400_Ce casting, a correspondence between the plastic behavior, Voce parameters, and presence of CHG on the fracture surface was observed. However, it should be noted that previous studies have reported that the fracture surface can exhibit a CHG density much higher than that found during metallographic investigation [12,29].

### 3.4. Micro-Computer Tomography (μCT)

Raw μCT images of cross and radial sections of the head of the tensile specimen C_2 from the GJS400_P casting are reported in Figure 17a,b, respectively, while raw μCT images of normal and radial sections of sample E_2 are reported in Figure 18a,b, respectively.

Because of the high density of CHG and eutectic cells in the GJS400_P casting, it was not possible to gather quantitative information about the graphitic agglomerate by using digital analysis of the metallographic micrographs in compliance with ASTM E2567-16a [44]. A similar problem was encountered for GJS400_Ce; however, the lack of a quantitative relationship between the metallographic results (Figure 4 and Figure 5) and the CHG observed on the fracture surfaces of the same samples (Figure 14, Figure 15 and Figure 16) suggested that volumetric information had to be gathered through μCT on defective GJS400. The measurements were focused on the samples E_2 and C_2 from the GJS400_P casting because the CHG density was very high in this casting, and furthermore, the CHG seemed to be concentrated in the eutectic cells, while big nodular agglomerates were embedded in the ferritic matrix only, which could facilitate the microstructure interpretation and parameters quantification.

The extremely fine size of the CHG within the eutectic cores made it challenging to identify CHG agglomerates and accurately determine the volume fraction of CHG using 3D imaging techniques due to the resolution limit (5 μm). However, μCT data analysis according to the procedure reported in the Experimental Section revealed two distinct volumes with different densities: *V_Ferrite_*, which refers to the denser volume of the ferritic matrix with coarse graphite agglomerates, and *V_Eutectic_*, which corresponds to the lower-density volume containing eutectic cells that consist of both fine CHG and coarse CHG at the boundaries. These measurements were conducted specifically for the GJS400_P casting, where the presence of CHG eutectic cells was significant, and most of the CHG was concentrated within the eutectic cells. On the other hand, in the GJS400_Ce, most of the CHG was embedded in the ferritic matrix, alongside significant amounts of simply degenerated graphite.

Figure 19b reports an image of the ferritic volume *V_Ferrite_* (in orange) embedding graphitic agglomerates, while the coral-like CHG in *V_Eutectic_* is still visible. In Figure 19a, the volumes *V_Ferrite_* (in orange) and *V_Eutectic_* of the eutectic cells (in yellow) of sample E_2 are depicted, while in Figure 20b, the volume *V_Eutectic_* only is reported. The quantitative results of the image analysis performed on the μCT reconstructions are listed in Table 3.

The quantitative μCT results summarized in Table 3 are consistent with the metallographic observations illustrated in Figure 2 and Figure 3. The largest size of the eutectic cells (volume *V_Ferrite_* with lower density) in C_2 is 1.37 mm, with a standard deviation of 0.69 mm, which is consistent with the wide eutectic cells reported in Figure 2a. The largest eutectic cell in sample E_2 was 0.30 mm, with a standard deviation of 0.12, which is consistent with the smaller eutectic cells reported in Figure 2b. Furthermore, in Figure 2, the size of the graphitic agglomerates in ferrite in the sample C_2 was bigger than in the sample E_2, resulting in a largest diameter of 80 μm, with a standard deviation of 25 μm, while in E_2, the largest diameter was 58 μm, with a standard deviation of 24 μm. The quantitative information gathered with μCT appeared reliable for the microstructure parameters comparable with the metallographic results. Finally, the volume fractions of eutectic cells were significantly higher in sample C_1 than in sample E_2, resulting in 34.1% of the total volume in sample C_2, while in sample E_2, the eutectic cell volume fraction was 27.1% of the total volume.

## 4. Discussion

### 4.1. Correlation between Tensile Behavior and Microstructure

The MAD and IAD plots for the tensile specimens from the defective GJS400_P and GJS400_Ce castings are shown in Figure 7 and Figure 8, respectively. For comparison, the plots for the sound GJS400_Y casting produced in Y-blocks are presented in Figure 9. In all the castings, the Voce parameters in MADs are distributed along lines, although there are significant differences. The best fitting lines for the defective materials have negative intercepts, namely, −18.6 in the GJS400_P casting and −8.8 in the GJS400_Ce casting, while the intercept is positive in the sound GJS400_Y casting, namely, +3.62 in Figure 9a. These results are consistent with previous investigations that have found negative intercepts in MADs for defective castings [19,20,30]. In IAD, the defective castings have data below the bisector lines that identify localized deformation or necking (see Figure 7b and Figure 8b), and the data are quite scattered. On the other hand, for the sound GJS400_Y casting in Figure 9b, the data are well above the bisector line with little scattering, indicating that the microstructure was quite homogeneous in the tested tensile specimens. Examining the tensile flow curves of the GJS400_P and GJS400_Ce castings in Figure 6, the defective materials flow curves display a wide variability of elongations to rupture and UTS, with quite constant YSs. In contrast, in sound castings, YS consistently increases with increasing UTS, while UTS and the elongations show low variability, as reported in [19,20,30].

Figure 9a shows a small spread of Voce parameters in MAD for the sound casting, which can be attributed to the relatively low variability of the sound microstructure, such as the grain size and amount of pearlite, as reported in previous studies [12,27,29]. However, Figure 7a and Figure 8a demonstrate a wide spread of Voce parameters in MAD for the GJS400_P and GJS400_Ce castings, respectively. This is likely due to the variability of the defects, including the typology and volumetric density of eutectic cells and CHG, which were the main types of defects in this study. The ductility behavior of these castings supports this conclusion, but a quantitative correlation has not been reported previously. Therefore, this study aimed to establish a quantitative relationship between defects and the trend in MAD. However, the metallographic observations made it difficult to quantify the CHG density in the defective castings. For instance, in GJS400_P, significant differences were found in the eutectic cell sizes between tensile specimens from the core of the prismatic heavy section (C_1 and C_2) and tensile specimens from the external surface (E_1 and E_2) in Figure 2 and Figure 3, respectively. Still, it was not possible to quantify CHG due to the complexity of the graphitic structure. Similarly, in the GJS400_Ce casting, Figure 4 shows a general presence of deteriorated graphitic agglomerates in tensile specimen Nr 17 (with the lowest Voce parameters in MAD, Figure 8a) and specimen Nr 20 (the highest Voce parameters in MAD, Figure 8a), while eutectic cells of CHG were found only in tensile specimen Nr 16. Thus, a single metallographic cut could not provide a reliable quantification of the CHG and eutectic cells, and a volumetric technique for microstructure characterization might have been more appropriate.

Fractography of the GJS400_Ce casting could help to correlate the presence of CHG in the tensile specimens Nr 20, 16, and 17, even if qualitatively only. In Table 4, the sample Nr 20,whichhad the lowest ductility (2.53%) and UTS (391.9 MPa), with the highest Voce parameters in MAD (Figure 8a) and the lowest data point in IAD (Figure 8b), presented a fracture surface completely covered in CHG (Figure 16). Sample Nr 17 showed the best ductility (17.85%) and UTS (461.4 MPa), with the lowest Voce parameters in MAD and the highest data point in IAD above the bisector line. Its fracture surface, as shown in Figure 14, had no evident CHG. In contrast, sample Nr 16, with tensile properties and Voce parameters in between, had a fracture surface that was only partially covered in CHG. These findings suggest that CHG has a direct effect on the plastic behavior and Voce parameters of the GJS400_Ce casting. However, in the GJS400_P casting, the fracture surfaces of all the tensile specimens from the core and external parts of the prismatic heavy section (Figure 11 and Figure 12 for C_1 and C_2, respectively, and Figure 13 and Figure 14 for E_1 and E_2, respectively) were completely covered in CHG, and thus, no conclusions could be drawn. However, it is essential to keep in mind that the information obtained from fractography might be more relevant for fracture behavior itself rather than the general plastic behavior of materials that concern the whole gauge volume. Fracture behavior can be influenced by local defects and metallurgical discontinuities. In fact, previous studies have reported that CHG (and graphitic agglomerates in general) are more numerous on the fracture surfaces than on the metallographic sections [8,29]. This can be attributed to the propensity of CHG (and defects in general) to act as sites of easy nucleation and propagation of cracks.

To obtain reliable quantitative information on the microstructure defects that could be related to plastic behavior, micro-computed tomography (μCT) measurements were conducted on the heads of tensile specimens C_2 and E_2 from the GJS400_P casting, as shown in Figure 17 and Figure 18, respectively. These samples were chosen for μCT measurements due to the high density of CHG in the GJS400_P casting and the significant differences in plastic behavior (Figure 6a) and Voce parameters in MAD and IAD (Figure 7) between the C_2 and E_2 specimens. μCT made it possible to distinguish between the metallic matrix and graphitic agglomerates based on their different densities (Figure 17 and Figure 18). However, due to the different mean densities of ferrite with embedded graphitic agglomerates and eutectic cells filled with CHG, an algorithm was developed with the aim of discriminating between two volumes: the first volume (*V_Ferrite_*) with a higher mean volumetric density, corresponding to the ferritic matrix with embedded large graphitic agglomerates, and the second volume (*V_Eutectic_*) with a lower mean volumetric density, corresponding to the eutectic cells filled with fine CHG. The quantitative results of the image analysis performed on the μCT reconstructions are presented in Table 3. Some of the μCT quantitative results in Table 3 were consistent with the microstructure metallographic observations of the C_2 and E_2 tensile specimens in Figure 2a,b, such as the mean diameter of the largest graphitic agglomerates in ferrite (80.0 μm in C_2 and 58 μm in E_2) and the sizes of the largest eutectic cells (1.37 mm in C_2 and 0.30 mm in E_2). Furthermore, in sample C_2 from the casting core, where the solidification cooling rate was lower, with a propensity to have more CHG, the eutectic cells occupied 34.1% of the total volume, while in sample E_2, the eutectic cells occupied only 27.1%. Since the Voce parameters in MAD reported in Figure 7a are related to the plastic behavior of tensile specimens, which is volumetric information concerning the specimen gauge volume, and since the information obtained through μCT was volumetric, it can be concluded that the significant differences in the tensile behaviors of the specimens from the core and the external part of the prismatic heavy section of the GJS400_P casting (Figure 6) and in Voce parameters in MAD and IAD (Figure 7) were due to the significantly different eutectic cell volume fractions with a high CHG density between the core and the external face of the prismatic heavy section.

In Figure 21a, the fractography results and the Voce parameters in MAD from the GJS400_Ce casting are reported on the same plot. The fracture surface of the tensile specimen Nr 17, with the lowest Voce parameters, presented a fracture surface with ferrite, nodular agglomerates, and no CHG. The fracture surface of the tensile specimen Nr 20, with the highest Voce parameters, shows a fracture surface completely covered in CHG (colored in brown for clarity), while the tensile specimen Nr 16, with Voce parameter values in between, presented a fracture surface partially covered in CHG. In Figure 21b, the μCT results and the Voce parameters in MAD from the GJS400_P casting show the correlation between the Voce parameters and the volume fractions (*V_Eutectic_*) of the eutectic cells that were dense in CHG: the tensile specimen E_2, with the lowest Voce parameters, had the lower eutectic cell density, with *V_Eutectic_* = 27.1%, while the tensile specimen C_2, with the highest Voce parameters, had the higher density of eutectic cells, with *V_Eutectic_* = 37.1%.

In conclusion, we have confirmed the correlation among tensile properties, Voce parameters in MAD and IAD, and defective microstructure in castings. This was accomplished qualitatively through fractography in the GJS400_Ce casting and quantitatively through μCT measurements in the GJS400_P casting. In the linear distribution of Voce parameters in MAD for defective materials, we found that the highest Voce parameters are consistent with highly defective castings, while the lowest Voce parameters are consistent with less defective ones. These findings align with experimental results that indicate higher Voce parameters are typical of flow curves with lower ductility and ultimate tensile strength (UTS), while lower Voce parameters correspond to higher elongations and UTS in Table 2.

### 4.2. Defects-Driven Plasticity (DDP)

Despite being naturally random, defects and metallurgical discontinuities do not affect the linearity of Voce parameters in defective castings in MADS, as shown in Figure 7a and Figure 8a. The main difference between sound and defective castings lies in the intercepts of the best fitting lines, which are positive for sound materials and negative for defective ones. This agrees with the physical interpretation of the Voce equation, where a positive intercept is expected for a sound material. In fact, the saturation stress is given by *σ_V_* = *Θ*_o_∙*ε*_c_, and since 1/*ε*_c_ =m∙*Θ*_o_ + C in MAD, it results in *σ_V_* =*Θ*_o_/(m∙*Θ*_o_ + C), in which *σ_V_* increases with increasing *Θ*_o_. The physical interpretation of the Voce equation indicates that *Θ*_o_ is inversely proportional to the characteristic length of the microstructure, such as the grain size and interlamellar distance. As such, an increase in *Θ*_o_ (refinement of the microstructure) is expected to result in an increase in *σ_V_* (tensile strength), which is a commonly accepted principle in plastic deformation theory. With defective castings like GJS400_P and GJS400_Ce, the best fitting lines with negative intercepts yield *σ_V_* = *Θ*_o_/(m∙*Θ*_o_ − C), resulting in a *σ_V_* (tensile strength) that decreases with increasing *Θ*_o_, which is in contrast with the physical meaning of the Voce equation. Figure 22 displays the *σ_V_* vs. *Θ*_o_ data for the GJS400_P and GJS400_Ce castings, as well as the sound GJS400_Y casting for comparison purposes. The range of *Θ*_o_ in Figure 22 is the same across all the castings for clarity. It is clear that the more defective the material, the more negative the intercept C becomes, and the more dramatic the inverse relationship between *σ_V_* and *Θ*_o_ becomes.

The Voce parameters in MADs show no distinct differences between sound and defective castings, as they lie similarly on lines. Surprisingly, a regular inverse relationship between the tensile strength and *Θ*_o_ was observed in defective castings, despite the random nature of the defects and metallurgical discontinuities. These characteristics led to the creation of the neologism “Defects-Driven Plasticity” (DDP) [19,20,30]. The random effects of defects and metallurgical discontinuities can be seen in the position of the Voce parameters on the best fitting line, where large Voce parameters correspond to highly defective materials with low ductility and UTS, while small Voce parameters correspond to less defective castings with higher ductility and UTS. Furthermore, high Voce parameters correspond to data below the bisector line in IAD, while small Voce parameters correspond to data closer to or above the bisector line in IAD. This investigation further supported these conclusions through quantitative measurements by establishing a correlation never reported before between Voce parameters in MAD and the volume fraction of eutectic cells dense with CHG in the GJS400_P casting, using μCT measurements.

### 4.3. Strain Hardening Considerations and a New Material Quality Index

Based on the considerations of the effects of defects and metallurgical discontinuities on tensile properties, from the data in IAD, a Specimen Quality Index (SQI) can be defined as:(3)$$\mathrm{SQI}=\frac{\varepsilon_{Rupture}}{\varepsilon_{Uniform}}$$

SQI refers to the single specimen, as the two parameters *ε_Rupture_* and *ε_Uniform_* come from the flow curve of the single tensile specimen. By averaging a statistical meaningful set of SQI values, a Material Quality Index (MQI) assessing the integrity of the casting can be found.

This approach has already been applied to Al castings produced by High-Pressure Die Casting (HPDC) [45], which bears similarities with the approach proposed by Caceres [46]. However, there are some key differences between the two approaches. First, in Equation (3), the denominator is the uniform strain calculated from the Voce parameters of the tensile flow curve itself, whereas in Caceres’ approach, the denominator is the uniform strain of an ideal defect-free material, referred to as the “major-defect-free specimen”. Second, the Hollomon equation was used to model the tensile flow curves in Caceres’ approach [46], so *ε_Uniform_* corresponds to the exponent *n* of the Hollomon constitutive equation that fits the tensile flow curve of the ideal Al casting free of defects, namely:(4)$$\sigma =K\cdot \varepsilon^{n}$$

Caceres’ approach has a drawback, as it is challenging to define the ideal Al casting since it would require a vast statistical analysis and depend heavily on the foundry best practices. This makes it difficult to establish a universally accepted “major-defect-free specimen”. The approach in Equation (3) also has some limitations. The reliability of the prediction of *ε_Uniform_* through Equation (3) depends on the “distance” between *ε_Uniform_* and *ε_Rupture_*, namely, the material ductility. Since *ε_Uniformi_* is calculated using the Voce parameters (*ε*_c_, *Θ*_o_, and *σ*_o_) obtained from fittings, the farther away *ε_Uniform_* is from *ε_Rupture_*, i.e., the poorer the ductility, the less reliable the prediction through Equation (3) may be.

However, the analysis of the linearity between the Voce parameters 1/*ε*_c_ and *Θ*_o_ can bring further insight into DDP. The linearity of the Voce parameters in MAD for any casting implies the existence of a microstructural parameter that is common to the casting from which the tensile specimens were cut, identifying the casting itself, regardless of the defects. Therefore, the best fitting line of the Voce data of defective castings in MAD can be expressed as:(5)$$\frac{1}{\varepsilon_{c}}=m\cdot \Theta_{o}-C$$
which can be written again as:(6)$$\frac{1}{\varepsilon_{c}}=\frac{\Theta_{o}}{\sigma_{P}}-\frac{\Theta_{P}}{\sigma_{P}}$$
where the stress *σ_P_* (MPa) and the strain hardening rate *Θ_P_* (MPa) are constant, derivedfrom m = 1/*σ_P_* and *C* = *Θ_P_*/*σ_P_*. Equation (6) can be inverted, resulting in:(7)$$\Theta_{P}=\Theta_{o}-\frac{1}{\varepsilon_{c}}\cdot \sigma_{P}$$

Since *Θ_o_* and 1/*ε_c_* identify a specific differential strain hardening curve of a set of data derived from a single casting, and thus from a specific tensile specimen, Equation (7) means that in the Kocks–Mecking plot (K-M plot), all the differential strain hardening curves of the tensile specimens from a single casting pass through a single point with coordinates (*σ_P_*;*Θ_P_*), herein after called the pivotal point. Figure 23 illustrates the K-M plots for the GJS400_P, GJS40_Ce, and GJS400_Y castings, where only three differential Voce equations of selected experimental tensile data are reported for clarity. In the K-M plots, the mean YS of the tensile flow curves of the single castings are indicated.

The pivotal point in the defective materials has a positive pivotal stress *σ_P_* and positive pivotal strain hardening strain rate *Θ_P_*. In the GJS400_P casting, the pivotal stress *σ_P_* is very close to the mean YS of the casting (and smaller than the saturation stresses *σ_V_*), so the ratio *σ_P_*/YS is 1.06, while in the GJS400_Ce casting, *σ_P_*/YS is 1.13. When the casting material is sound, the pivotal point is not real, as the pivotal stress *σ_P_* is positive and larger than the saturation stresses *σ_V_*, while the pivotal strain hardening rate *Θ_P_* is negative. In Figure 24, the analysis of the differential data from the GJS400_Y casting is reported, putting in evidence that the selected strain hardening best fitting lines converge to the pivotal point (714.7; −2597.0) MPa; now the “distance” between the mean YS and pivotal stress *σ_P_* is larger, as the ratio *σ_P_*/YS is 2.47.

It is noteworthy that the above findings are related to the negativity or positivity of the intercept C of the best fitting lines of the Voce parameters in MADs. Indeed, some considerations can be drawn:in sound materials, *σ_P_* is higher than the saturation stresses *σ_V_* of the tensile flow curves, and, as a consequence, the pivotal strain hardening rate *Θ_P_* is negative (non-real), as indicated in Figure 24b;in defective materials, the pivotal stress *σ_P_* is between the mean YS and the saturation stresses *σ_V_*, so that the strain hardening rate *Θ_P_* is positive (real), as shown in Figure 23a,b;as the material defectiveness increases, the mean yield strength (YS) and pivot stress (*σ_P_*) become closer, causing the ratio of *σ_P_*/YS to approach unity in very defective castings;with improving casting integrity, the ratio *σ_P_*/YS increases.

According to the differential form of the Voce equation, the parameter *Θ_o_* is related to hardening, while the slope 1/*ε_c_* is related to the softening of materials. In defective materials, a correlation between the parameters 1/*ε_c_* and *Θ_o_* has been observed, indicating their relation to the presence and type of defects, rather than to the physical meaning of the Voce equation. While the physical interpretation of the Voce equation remains an open issue, this finding is a significant step forward in understanding Defects-Driven Plasticity. It reveals the existence of a pivotal point common to all the differential tensile data, which identifies the casting and moves the focus from the intercept of the best fitting lines of the Voce parameters in MADs to the pivotal point (*σ_P_*;*Θ_P_*).

However, a practical advantage of this new finding can be exploited. Though the real meaning of the correlation between 1/*ε_c_* and *Θ_o_* is not available at the moment, the ratio *σ_P_*/YS seemed to be an indication of how the material is defective, as the ratio is correlated to the intercept: in a highly defective material with a negative intercept (C < 0), the ratio *σ_P_*/YS is close to unity, while with C closer to 0 and even positive (sound casting), the ratio *σ_P_*/YS increases progressively. Therefore, a new MQI of the castings can be proposed, based on the “distance” between the mean YS and the pivotal stress *σ_P_*, as follows:(8)$${\mathrm{MQI}}_{Pivot}=\ln\left(\frac{\sigma_{P}}{YS_{mean}}\right)$$

Looking at the investigated castings in Figure 23 and Figure 24, Equation (8) results in:GJS400_Y −MQI*_Pivot_* = 0.97;
GJS400_Ce −MQI*_Pivot_* = 0.12;
GJS400_P −MQI*_Pivot_* = 0.06

The new MQI can facilitate an integrity assessment of DI grades with comparable microstructures, such as ferritic, ferritic–pearlitic, pearlitic, isothermed ferritic–pearlitic or ausferritic, compacted ferritic, etc. The comparison can be made between similar chemical compositions, but different cooling rates or different chemical compositions with constant production routes. The advantages of using MQI*_Pivot_* over the one based on Equation (3) are manifold. First, YS is a readily measurable engineering parameter and tends to be constant with defective casting materials, becoming a microstructure casting parameter. Second, the pivotal stress *σ_P_* is a genuine microstructure parameter that is related to the individual casting.

## 5. Conclusions

The present study investigated the microstructure and tensile behavior of two heavy section castings with chemical compositions typical of GJS400, and compared them to a sound GJS400 produced through Y-blocks that had been previously investigated. The microstructures of the castings were fully ferritic, with varying degrees of degenerated and Chunky Graphite (CHG), and were analyzed through conventional metallography, fractography, and micro-Computer Tomography (μCT) to quantify the volume fractions of eutectic cells with CHG.

The findings of this study shed light on Defects-Driven Plasticity (DDP), which refers to the unexpected regular plastic behavior of materials due to defects and metallurgical discontinuities, resulting in the linearity of Voce parameters in Matrix Assessment Diagrams (MADs), where the intercept of the best fitting line of the Voce parameters is negative, which contradicts the physical meaning of the Voce equation parameters. Specifically, the study revealed qualitatively, through fractography, and quantitatively, through μCT measurements, that:The lowest positions of the Voce parameters in the linear dataset correspond to the least defective microstructure, having also the longest elongations to rupture and the highest UTS;The highest positions of the Voce parameters in the linear dataset correspond to the most defective microstructure, also displaying the shortest elongations to rupture and the smallest UTS.

A new finding was reported, resulting in a step forward into the comprehension of DDP:3.The linearity of the Voce parameters in MAD translates into the existence of a pivotal point belonging to all the experimental Voce differential curves, namely, (*σ_P_*;*Θ_P_*);4.In sound materials, *σ_P_* is higher than the saturation stresses *σ_V_* of the tensile flow curves, and, as a consequence, the pivotal strain hardening rate *Θ_P_* is negative (non-real);5.In defective materials, the pivotal stress *σ_P_* is between the mean YS and the saturation stresses *σ_V_*, so that the strain hardening rate *Θ_P_* is positive (real);6.As the material defectiveness increases, the mean yield strength (YS) and pivot stress (*σ_P_*) become closer, causing the ratio of *σ_P_*/YS to approach unity in defective castings, while with improving casting integrity, the ratio *σ_P_*/YS increases well above the unity.

These findings can be exploited to assess the integrity of castings: a new MQI of castings is proposed, based on the “distance” between the mean YS and the pivotal stress *σ_P_*. MQI is close to 0 for highly defective castings (GJS400_P), and increases with the increasing integrity of the casting, achieving, for instance, 0.97 for the investigated sound GJS400_Y.

The rationalization of DDP is still an ongoing research topic. However, the discovery of the pivotal point (*σ_P_*;*Θ_P_*) represents an important step forward in understanding DDP. This knowledge could potentially lead to its proper application in the integrity assessment of castings.

## Figures and Tables

**Figure 1 materials-16-03748-f001:**
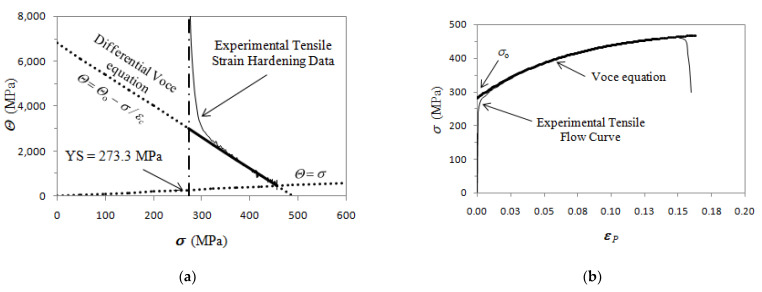
Voce equation fitting procedure for a typical tensile flow curve of GJS400: (**a**) tensile strain hardening data in K-M plot; (**b**) tensile flow curve *σ* vs. *ε_P_*. The Yield Stress (YS) is reported, while *Θ* = *σ* corresponds to the Considére’s criterion identifying the uniform strain.

**Figure 2 materials-16-03748-f002:**
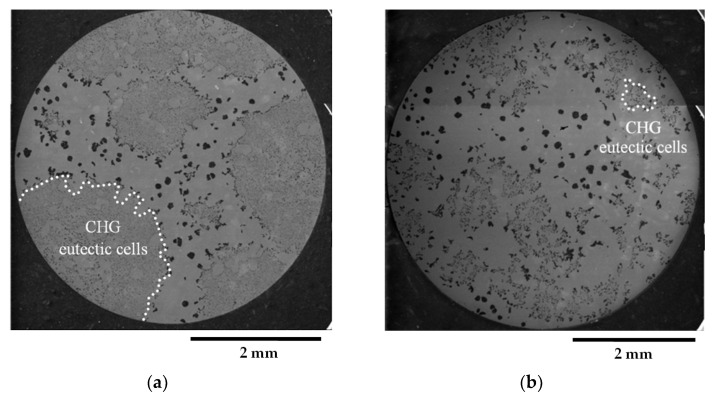
Representative SEM micrographs (BEI)of GJS400_P casting microstructure: (**a**) tensile specimen head C_2 and (**b**) tensile specimen head E_2. Selected eutectic cells are surrounded with dotted lines for clarity.

**Figure 3 materials-16-03748-f003:**
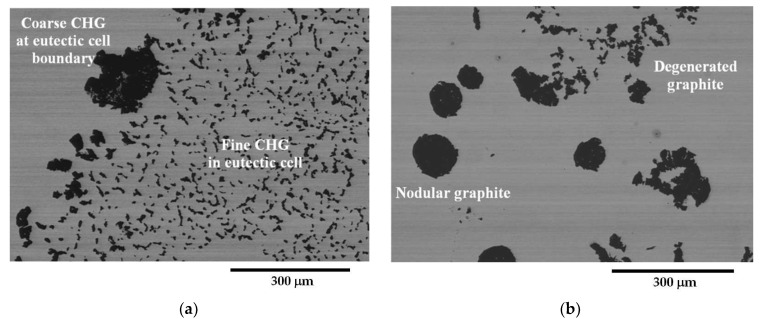
Representative SEM micrographs (BEI)showing the details at higher magnifications of graphite microstructure inGJS400_P casting: (**a**) tensile specimen head C_2 and (**b**) tensile specimen head E_2.

**Figure 4 materials-16-03748-f004:**
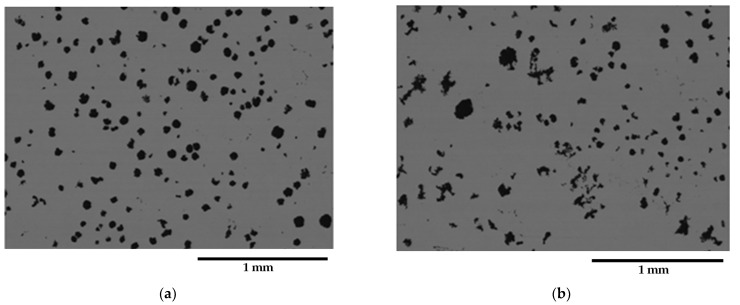
Representative SEM micrographs (BEI) of GJS400_Ce casting showing the graphite microstructure from: (**a**) tensile specimen head in sample Nr 17 and (**b**) tensile specimen head in sample Nr 20.

**Figure 5 materials-16-03748-f005:**
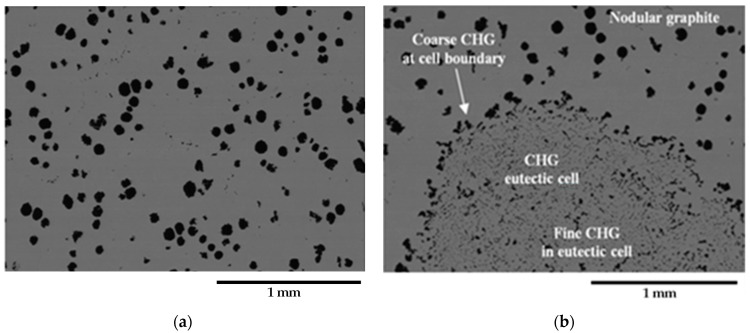
Representative SEM micrographs (BEI) of GJS400_Ce casting showing the graphite microstructure of the tensile specimen head of sample Nr 16: (**a**) nodular graphitic agglomerate, (**b**) fine CHG in eutectic cell, and coarse CHG at the eutectic cell boundaries.

**Figure 6 materials-16-03748-f006:**
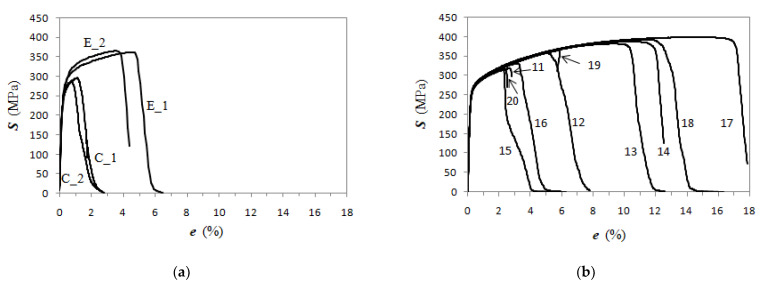
Engineering tensile flow curves of tensile specimens from castings: (**a**) GJS400_P; (**b**) GJS400_Ce.

**Figure 7 materials-16-03748-f007:**
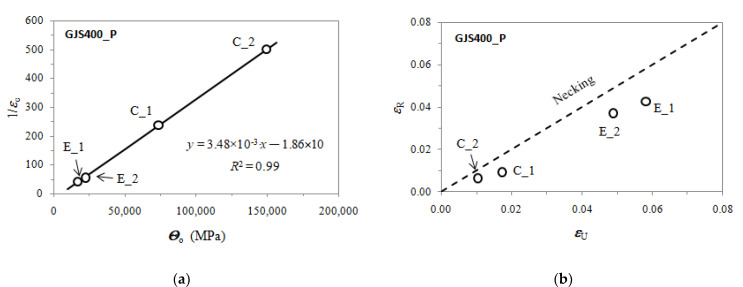
Voce parameters from GJS400_P casting: (**a**) MAD; (**b**) IAD.

**Figure 8 materials-16-03748-f008:**
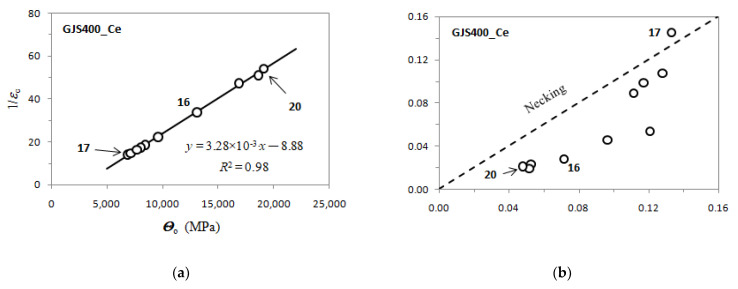
Voce parameters from GJS400_Ce casting: (**a**) MAD; (**b**) IAD.

**Figure 9 materials-16-03748-f009:**
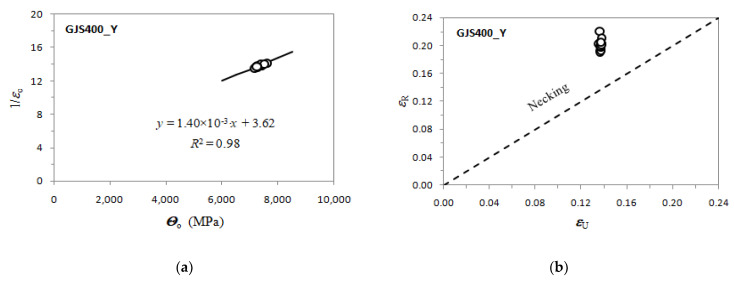
Voce parameters from GJS400_Y casting: (**a**) MAD; (**b**) IAD.

**Figure 10 materials-16-03748-f010:**
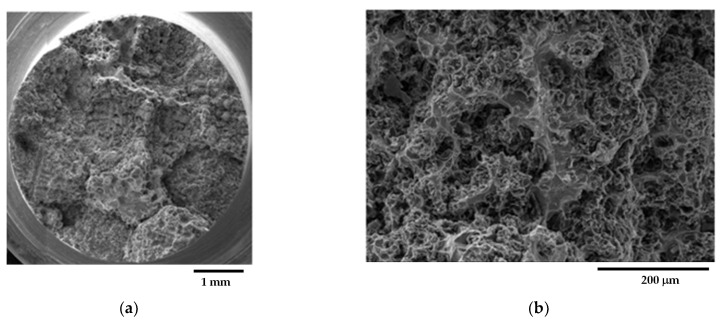
Representative SEM micrographs (SEI) at different magnifications showing the fracture surface of tensile specimen C_1 from GJS400_P casting: (**a**) general view; (**b**) ductile fracture region with fine dimples and CHG; (**c**) final fracture region with cleavage facets, nodules, and shrinkage porosity; (**d**) details of shrinkage porosity (enclosed in red) at higher magnification.

**Figure 11 materials-16-03748-f011:**
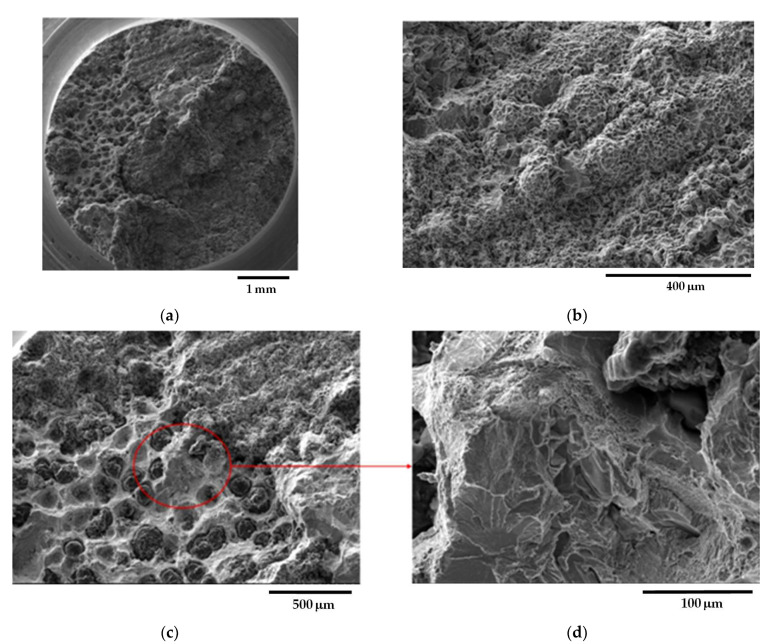
Representative SEM micrographs (SEI) at different magnifications showing the fracture surface of tensile specimen C_2 from GJS400_P casting: (**a**) general view; (**b**) ductile fracture region with small dimples and CHG; (**c**) final fracture region with cleavage facets and graphitic nodules; (**d**) details of cleavage facets.

**Figure 12 materials-16-03748-f012:**
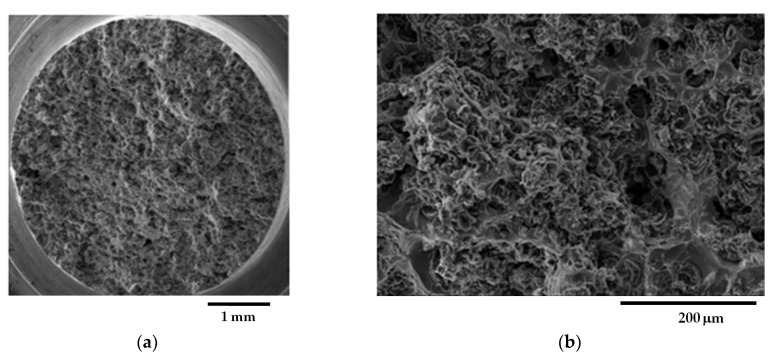
Representative SEM micrographs (SEI) at different magnifications showing the fracture surface of tensile specimen E_1 from GJS400_P casting: (**a**) general view; (**b**) ductile fracture region with CHG and little cleavage facets with nodules (below right).

**Figure 13 materials-16-03748-f013:**
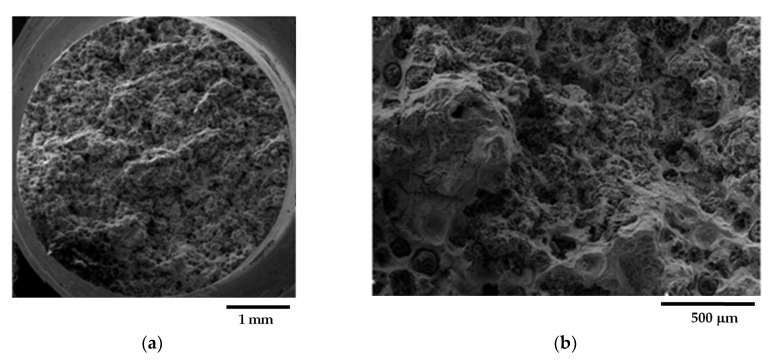
Representative SEM micrographs (SEI) at different magnifications showing the fracture surface of tensile specimen E_2 from GJS400_P casting: (**a**) general view; (**b**) ductile fracture region with CHG and last fracture region with cleavage facets and nodules (below left).

**Figure 14 materials-16-03748-f014:**
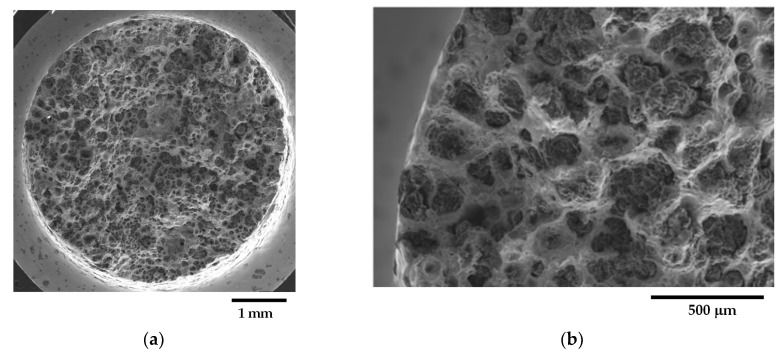
Representative SEM micrographs (SEI) at different magnifications showing the fracture surface of tensile specimen Nr 17: (**a**) general view; (**b**) ductile fracture region and last region to fracture with cleavage facets and nodules.

**Figure 15 materials-16-03748-f015:**
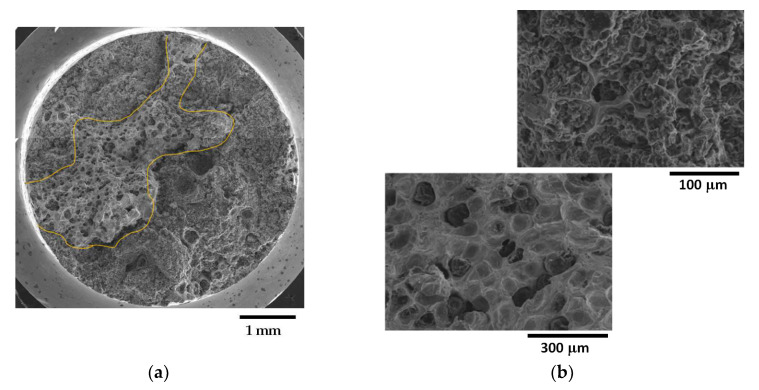
Representative SEM micrographs (SEI) at different magnifications showing the fracture surface of tensile specimen Nr 16: (**a**) general view with ferritic region and nodule (underlined in yellow) and regions cover in CHG; (**b**) details at higher magnifications of regions cover in CHG (micrographs above), and regions with ferritic matrix and nodules (micrographs below).

**Figure 16 materials-16-03748-f016:**
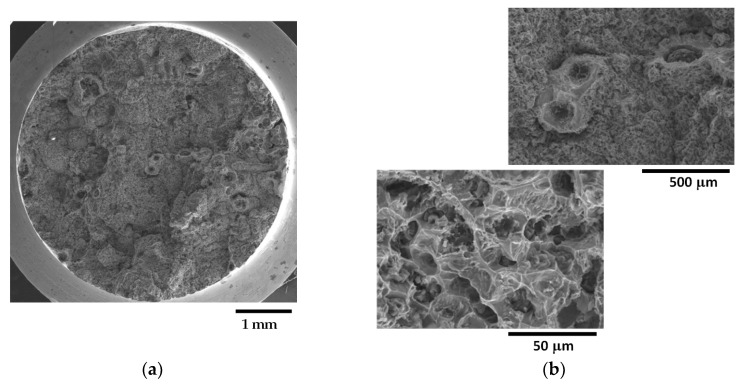
Representative SEM micrographs (SEI) at different magnifications showing the fracture surface of tensile specimen Nr 20: (**a**) general view; (**b**) fracture surface covered in CHG (micrographs above) and last fracture region with cleavage facets and nodules (micrograph below).

**Figure 17 materials-16-03748-f017:**
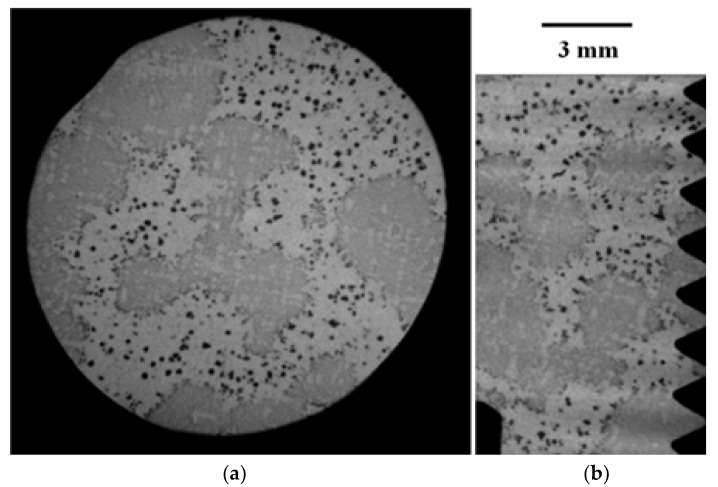
Representative raw μCT images of sections of the threaded head of the tensile specimen C_2 from GJS400_P casting: (**a**) cross section; (**b**) axial section.

**Figure 18 materials-16-03748-f018:**
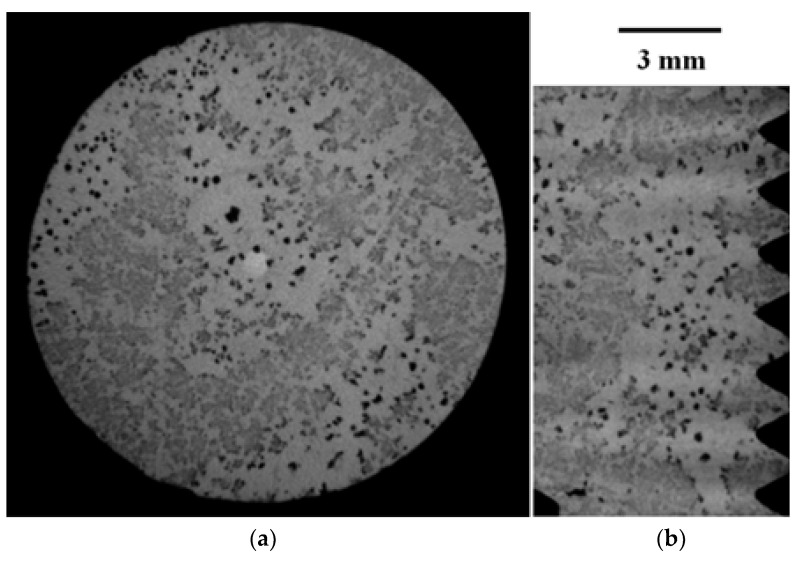
Representative μCT images of sections of the threaded head of the tensile specimen E_2 from GJS400_P casting: (**a**) cross section; (**b**) axial section.

**Figure 19 materials-16-03748-f019:**
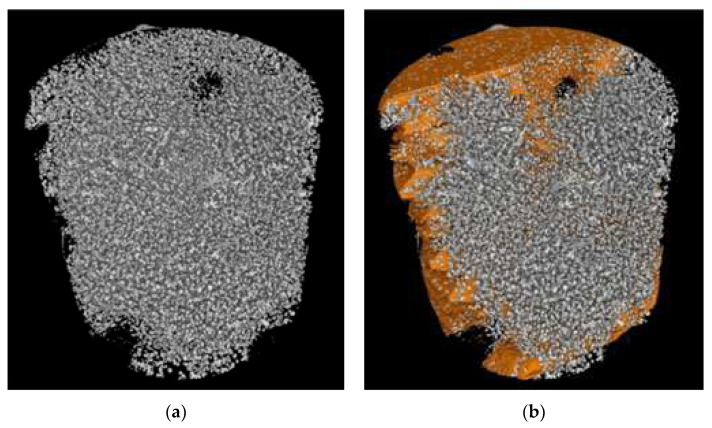
Representative μCT reconstructions: (**a**) graphitic agglomerates in sample E_2; (**b**) ferritic matrix *V_Ferrite_* (in orange) with embedded graphitic agglomerates, while the coral-like CHG in *V_Eutectic_* is still visible.

**Figure 20 materials-16-03748-f020:**
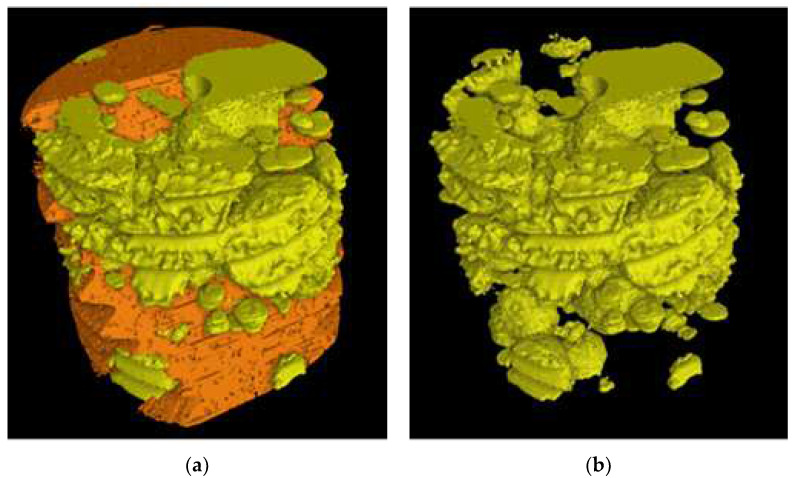
Representative μCT reconstructions: (**a**) ferritic matrix *V_Ferrite_* (in orange) with embedded graphitic agglomerates, while the eutectic cells (volume *V_Eutectic_* in yellow) are visible; (**b**) the volume *V_Eutectic_* is only observable.

**Figure 21 materials-16-03748-f021:**
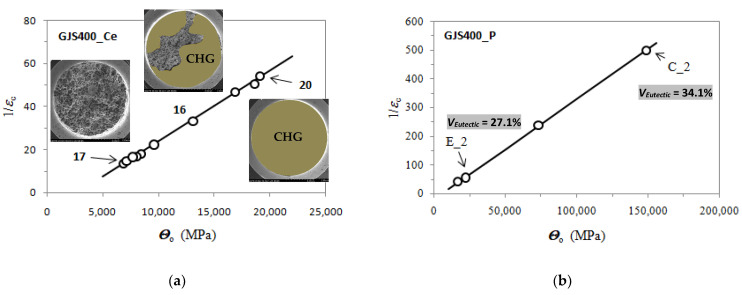
Correlation between Voce parameters in MAD and microstructure observation: (**a**) CHG areas (in brown) on fracture surface in GJS400_Ce casting; (**b**) eutectic volume fractions (*V_Eutectic_*) calculated with μCT technique in GJS400_P.

**Figure 22 materials-16-03748-f022:**
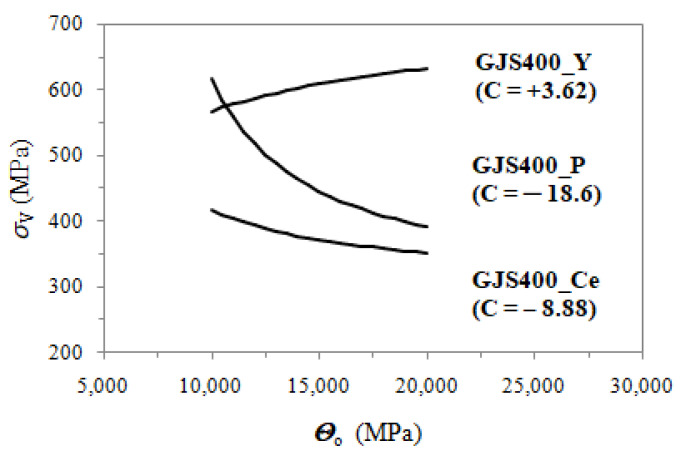
*σ_V_* vs. *Θ*_o_ data for the defective GJS400_P and GJS400_Ce castings; the *σ_V_* vs. *Θ*_o_ data of the sound GJS400_Y casting is reported for comparison purposes.

**Figure 23 materials-16-03748-f023:**
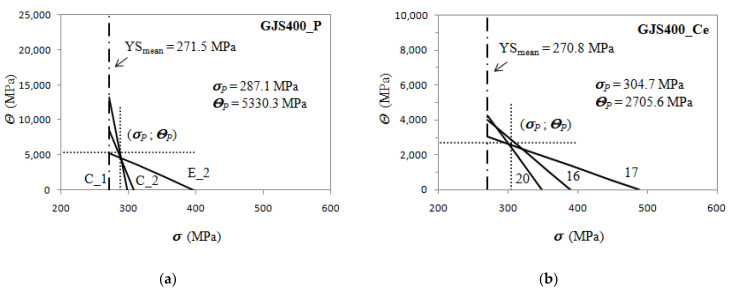
*Θ* vs. *σ* data for the defective castings: (**a**) GJS400_P; (**b**) GJS400_Ce.

**Figure 24 materials-16-03748-f024:**
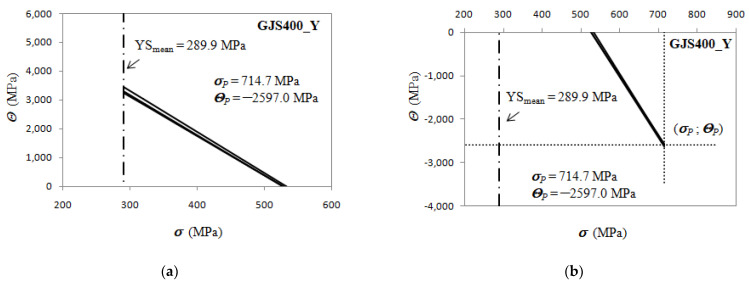
*Θ* vs. *σ* data for the sound GJS400_Y casting: (**a**) positive quadrant of *Θ* vs. *σ* plot; (**b**) negative quadrant to put in evidence the pivotal point.

**Table 1 materials-16-03748-t001:** Chemical composition of the heavy sections reported in wt%.

Code	C	Si	Mn	P	S	Mg	Cr	Ce	C_Eq_ *
GJS400_P	3.60	2.29	0.08	0.039	0.011	0.053	0.005	-	4.38
GJS400_Ce	3.75	2.32	0.16	-	0.009	0.046	0.040	0.006	4.53
GJS400_Y	3.63	2.45	0.13	0.038	0.043	0.046	0.023	-	4.46

* Carbon equivalent: C_Eq_% = C% + (Si% + P%)/3.

**Table 2 materials-16-03748-t002:** Tensile mechanical properties of specimens from GJS400_P and GJS400_Ce castings.

Casting	Tensile Specimen In-House Code	E (GPa)	YS (MPa)	UTS (MPa)	Elongation to Rupture (%)
GJS400_P	C_1	185.6	260.4	295.5	2.62
	C_2	193.5	261.8	286.8	2.80
	E_1	168.8	282.6	361.5	6.47
	E_2	167.1	281.0	366.0	4.37
GJS400_Ce	11	151.4	269.4	326.3	2.82
	12	169.6	272.5	376.5	7.80
	13	154.8	274.9	418.8	12.63
	14	157.6	270.6	429.3	12.53
	15	163.3	271.4	329.2	4.44
	16	174.7	269.1	341.3	5.95
	17	162.0	273.3	461.4	17.85
	18	159.6	269.5	438.1	14.69
	19	168.0	269.6	388.3	5.42
	20	162.7	267.4	321.9	2.53

**Table 3 materials-16-03748-t003:** Quantitative information gathered through analysis of μCT measurements.

Sample In-House Code	C_2	E_2
Graphitic agglomerates volume fraction in ferrite (%)	6.3	6.1
*V_Ferrite_* (ferrite + graphitic agglomerates) (%)	65.9	72.9
*V_Eutectic_* (eutectic cells with CHG) (%)	34.1	27.1
Largest eutectic cell diameter (mm)	1.37	0.30
Standard deviation of the largest eutectic cell diameter (mm)	0.69	0.12
Largest ferritic volume diameter (mm)	1.37	1.04
Standard deviation of the largest ferritic volume diameter (mm)	0.69	0.57
Largest graphitic agglomerate diameter in *V_Ferrite_* (mm)	0.080	0.058
Standard deviation of the largest graphitic agglomerate diameter in *V_Ferrite_* (mm)	0.025	0.024

**Table 4 materials-16-03748-t004:** Voce parameters of specimens from castings GJS400_Prism and GJS400_Ce.

Casting	Tensile Specimen In-House Code	*Θ*_o_ (MPa)	1*/ε*_c_	*σ*_V_ (MPa)	*σ*_o_ (Mpa)	*ε_Crit_*	*ε_Unif_*
GJS400_P	C_1	73,282.1	238.2	307.6	231.5	0.009	0.017
	C_2	148,939.4	499.8	298.0	201.1	0.006	0.010
	E_1	16,841.8	42.5	396.3	288.1	0.043	0.058
	E_2	22,435.2	56.2	399.2	291.3	0.037	0.049
GJS400_Ce	11	16,818.1	47.1	356.8	267.5	0.024	0.053
	12	9619.9	22.2	434.8	274.8	0.046	0.097
	13	8366.0	18.3	456.1	274.9	0.090	0.111
	14	7980.3	17.1	465.3	273.6	0.099	0.117
	15	18,563.8	50.8	365.2	267.4	0.020	0.052
	16	13,097.1	33.6	389.6	266.9	0.029	0.071
	17	6790.2	13.9	488.6	281.4	0.146	0.133
	18	7156.1	15.0	478.3	274.6	0.108	0.128
	19	7710.4	16.5	467.7	272.5	0.054	0.121
	20	19,017.7	54.4	349.4	265.2	0.021	0.048

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
