# Peer review of "An Insight into the Defects-Driven Plasticity in Ductile Cast Irons"

_materials, 2023, doi:10.3390/ma16103748_

Round 1
Reviewer 1 Report
1. Induction: In the analysis is missing the testing method hot tensile static test for observation of defects formation during material transformation from liquid to a solid state discussed e.g. [P. Prislupcak, et al: Effect of Austenitization Temperature on Hot Ductility of C-Mn-Al HSLA Steel, Materials 2022, 15, 922. https://doi.org/10.3390/ma15030922]
2. The article at the present form has not character of a scientific paper from point of view his structure. E.g.: the content is mixing experimental methods with results together. The authors must divide the content very preciously into chapters: Materials and experimental methods, Results, Discussion.
3. Tab.1: The chemical compositions are characterized by local analyses or melt analyses? The content of other chemical elements such as Cu, Ni, Cr, Ti and gases O2, N2 are missing.
4. The definition of the shape and dimensions of the samples for the static tensile test is missing.
5. Ce has a high affinity to the S. What was the reason for adding Ce to the tested material?
6. Eq.(1a) and Eq.(1b): an explanation of the variables in these equations are missing.
7. Θ0 - authors described this variable as "dislocation multiplication factor". But Θ0=dσ/dε [MPa] is characterized as strain hardening which depend on different deformation mechanisms. What kind of deformation mechanism was observed at the material testing?
8. Conclusions: The nature of this chapter is very descriptive without defining the material that showed the best properties depending on the test methods.
Author Response
Reply to Reviewer Nr 1
Dear Reviewer,
thanks a lot for your comments and suggestions that have helped us to improve the quality of our paper.
Below you find the detailed reply to your comments and the actions we took to follow your suggestions. In the MARKED version of the paper you can read the text that concerns our actions underlined in green.
Kind regards
Giuliano Angella, PhD
…………………………………………………………………
Reviewer’s comment 1. Introduction: In the analysis is missing the testing method hot tensile static test for observation of defects formation during material transformation from liquid to a solid state discussed e.g. [P. Prislupcak, et al: Effect of Austenitization Temperature on Hot Ductility of C-Mn-Al HSLA Steel, Materials 2022, 15, 922. https://doi.org/10.3390/ma15030922]
Action – In Introduction we wrote statements about the relevance of hot tensile testing in the investigation of the transformation from liquid to solid state that foster the crack formation during solidifications in steels. The new statement with the suggested reference and other relevant ones are underlined in green in the MARKED paper at page 1:
- INTRODUCTION
Defects and metallurgical discontinuities form during metallic alloys solidification, and can affect significantly the magnitude and the variability of mechanical properties. In castings of Ductile Irons (DIs), for instance, main defects and discontinuities can be degenerated graphite agglomerates, dross, inclusions, gas and solidification shrinkage porosities etc., which can have detrimental impact on several mechanical properties, such as room temperature tensile strength, ductility, fatigue resistance, and fracture toughness [1-7]. In continuous casting production of steels, for instance, the metallurgical discontinuities like cracks forming during solidification at the slab surface influence appreciably the material quality resulting from the subsequent slab straightening [8,9]. In this case, hot tensile strength and ductility investigations have been reported to be effective tools to investigate the transformations from liquid to solid state that can foster the crack formation during solidification [10,11].
Reviewer’s comment 2. The article at the present form has not character of a scientific paper from point of view his structure. E.g.: the content is mixing experimental methods with results together. The authors must divide the content very preciously into chapters: Materials and experimental methods, Results, Discussion.
Action – The structure of the paper was significantly changed according to the Reviewer’s suggestions: so the order of the paragraphs, the figures and the tensile results was changed, giving to the paper a clearer structure, easer to read.
Reviewer’s comment 3. Tab.1: The chemical compositions are characterized by local analyses or melt analyses? The content of other chemical elements such as Cu, Ni, Cr, Ti and gases O2, N2 are missing.
Action – In Table 1 the chemical composition of the most relevant chemical elements is reported, whilst other elements like Cu, Ni and Ti were checked but their contents resulted < 0.01wt%. The chemical technique used (Optical Emission Spectroscopy) and the procedure was reported in the text at page 4 of the MARKED_paper, whilst O2 and N2 gases were not measured.
2.1. Materials and microstructure characterization
Two heavy sections with GJS400 grade chemical compositions were investigated: chemical compositions of the relevant elements are reported in Table 1. The chemical analyzes were carried out using an Optical Emission Spectrometer (OES) on DI tablets obtained by rapid solidification of the melts, in order to have white cast irons with no graphite that would have interfered with OES measurements. Also other elements, like Cu, Ni and Ti, were checked according to standard cast irons foundry practice, resulting in quantities < 0.01wt%, whilst O and N contents were not measured.
Reviewer’s comment 4. The definition of the shape and dimensions of the samples for the static tensile test is missing.
Action – Tensile specimen geometry and tensile testing procedure are complying with ASTM E8-E8M, and the information are reported at the end of paragraph EXPERIMENTAL, page 5, of the MARKED_paper, underlined in green:
The DIs were tensile tested: round tensile specimens with initial gauge length lo = 28.0 mm and diameter do = 5.6 mm were machined off and tensile tested complying with standard ASTM E8/E8M – 11 [3442] at a constant strain rate of 10-4 s-1. Engineering stress S = F/Ao, where F is the applied force and Ao is the initial gauge cross sectional area, and elongation e = (l – lo)/lo, where l is the instantaneous gauge length, were transformed into true stress s = S∙( 1 + e) and true strain e = ln( 1 + e). Only the true plastic strain ep was considered for strain hardening analysis, where ep = e – eel, with eel = s/E, and E the elastic Young's modulus.
Reviewer’s comment 5. Ce has a high affinity to the S. What was the reason for adding Ce to the tested material?
Action – Ce was added with inoculation with the intention of degenerating the nodular graphite and to foster the formation of chunky graphite. Unfortunately we were not enough clear on the reasons why we added Ce, and now this should have been clarified at page 4 of the MARKED_paper:
The second GJS400 (code GJS400_Ce) had some Ce coming from adding 0.2% of Ce inoculation with the intention of degenerating the nodular graphite and foster the formation of CHG, and was produced into a cylindrical block of 300 mm diameter and 520 mm height.
Reviewer’s comment 6. Eq.(1a) and Eq.(1b): an explanation of the variables in these equations are missing.
Action – Sorry, the variables were not explained; now they should be defined correctly at page 2 of the MARKED_paper:
The analysis of the tensile strain hardening at room temperature in defective materials using the dislocation-density-related constitutive equation of Voce, which is based on physical principles, can provide valuable information regarding the defective nature of DIs [15,16,2719,20,31]. Voce constitutive equation is:
where s and eP are the true stress and plastic strain, respectively; sV is the saturation stress or maximum stress; eC is the critical plastic strain that defines the rate with which the saturation stress is achieved; and so is the back-extrapolated stress to zero plastic strain, close to yield stress.
Reviewer’s comment 7. Θ0 - authors described this variable as "dislocation multiplication factor". But Θ0=dσ/dε [MPa] is characterized as strain hardening which depend on different deformation mechanisms. What kind of deformation mechanism was observed at the material testing?
Action – The reviewer is correct, the definition of the Voce variables in the differential form of Voce equation was not detailed. We did not carry out any microscopic observation that was not the aim of the investigation, but we assumed that glide was the relevant dislocation motion mechanism, and cross slip could be relevant in dynamic recovery according to [Kocks, U.F.; Mecking, H. Prog. Mater Sci. 2003, 48, 171–273]. Now the variables Θ0 and 1/ ec should be defined correctly, detailing the basic principles of dislocation dynamics at page 2 of the MARKED_paper:
Voce parameters in Eq. 1a can be found from the analysis of the tensile strain hardening data through the differential form of the constitutive Voce equation, defined as:
Equation 1b has physical meaning, as 1/ec is the softening term related to dynamic recovery that comes into dislocation annihilation and formation of low energy dislocation structures [3532], and Qo is the dislocation multiplication factor because of dislocation storage at internal deformation structures and grain boundaries, and according to Kocks-Mecking model Qo is inversely related to dislocation cells dimensions, grain size and other obstacles to dislocation motion.
Reviewer’s comment 8. Conclusions: The nature of this chapter is very descriptive without defining the material that showed the best properties depending on the test methods.
Action – We are sorry to realize that the conclusions were not clearly underlined. So the number of pointed statements was reduced in CONCLUSIONS in page 25 of the MARKED_paper, in order to make clear the achievements of this investigation:
- The correlation between extent of defects in the defective materials and position in the linear distribution of Voce parameters in the Matrix Assessment Diagrams;
- The finding of a pivotal point in the Kocks-Mecking plot that can give a new important insight in the defects Driven Plasticity.
- CONCLUSIONS
…..
The findings of this study shed light on Defect Driven Plasticity (DDP), which refers to the unexpected regular plastic behavior of materials due to defects and metallurgical discontinuities, resulting in the linearity of Voce parameters in Matrix Assessment Diagrams (MADs) where the intercept of the best fitting line of Voce parameters is negative, which contradicts the physical meaning of the Voce equation parameters. Specifically, the study revealed qualitatively through fractography and quantitatively through mCT measurements that:
- The lowest positions of the Voce parameters in the linear data set correspond to the least defective microstructure, having also the shortest longest elongations to rupture and smallest highest UTS;
- The highest positions of the Voce parameters in the linear data set correspond to the most defective microstructure, also displaying the highest shortest elongations to rupture and smallest UTS.
A new additional finding was reported, resulting in a step forward into the comprehension of DDP:
- The linearity of the Voce parameters in MAD translates in the existence of a pivotal point belonging to all the experimental Voce differential curves, namely, (sP;QP);
- In sound materials, sP is higher than the saturation stresses sV of the tensile flow curves, and, as a consequence, the pivotal strain hardening rate QP is negative (non-real);
- In defective materials, the pivotal stress sP is between the mean YS and the saturation stresses sV, so that the strain hardening rate QP is positive (real);
- As material defectiveness increases, the mean yield strength (YS) and pivot stress (sP) become closer, causing the ratio of sP/YS to approach unity in defective castings; whilst with improving casting integrity, the ratio sP/YS increases well above the unity.
......................................
We hope that now the achievements are more clearly reported. Thanks to the Reviewer for his/her notably contribution to improve the conclusion exposition.
Reviewer 2 Report
This study investigated the microstructure and tensile behavior of two heavy section castings that had chemical compositions typical of GJS400. Frankly speaking, I cannot recommend its acceptance. This is a quite routine work; this is more like an experimental report rather than academic paper. The author should try give more discussions about the current data and find some interesting phenomenon and mechanism. In addition, there are too many SEM micrographs and they are not wll discussed.
Moderate editing of English language.
Author Response
Dear Reviewer,
Thanks a lot for your comments that have helped us to improve the quality of our paper. We are sorry that the Reviewer did not appreciate the novelty reported in our paper. Thanks to the suggestions of all Reviewers we think that now the quality of the paper improved significantly putting in evidence the achievement of our investigation.
Below you find the detailed reply to your comments and the actions we took to improve the paper. In the MARKED version of the paper you can read the text that concerns your suggestions underlined in grey.
Kind regards
Giuliano Angella, PhD
…………………………………………………………………
Reviewer’s comment 1. This study investigated the microstructure and tensile behavior of two heavy section castings that had chemical compositions typical of GJS400. Frankly speaking, I cannot recommend its acceptance. This is a quite routine work; this is more like an experimental report rather than academic paper.
Action – We are sorry that the paper did not touch the Reviewer’s interest. Indeed, in the investigations we reported some novel results never reported before:
- We proved the correlation between defects in defective materials and the position in the linear distribution of Voce parameters in the Matrix Assessment Diagrams;
- We explained in details the Defects Driven Plasticity (DDP), which has been introduced in other papers, but it has been treated deeply for the first time in this paper;
- The findings of a pivotal point in the Kocks-Mecking plots for each casting, which gives a new important insight in the Defects Driven Plasticity to move forward in the comprehension of DDP.
So, as it was suggested correctly by the Reviewer Nr 3, there were no summarizing plots of the achievements, so we introduced a summarizing figure (Figure 21 at page 21 of the MARKED_paper) and an explaining text of the figure:
In Figure 21a the fractography results and the Voce parameters in MAD from the GJS400_Ce casting are reported on the same plot. The fracture surface of the tensile specimen Nr 17 with the lowest Voce parameters presented a fracture surface with ferrite, nodular agglomerates and no CHG. The fracture surface of the tensile specimen Nr 20 with the highest Voce parameters shown a fracture surface completely covered in CHG (colored in brown for clarity purposes), whilst the tensile specimen Nr 16 with Voce parameter values in between, presented a fracture surface partially covered in CHG. In Figure 21b the mCT results and the Voce parameters in MAD from the GJS400_P casting shown the correlation between Voce parameters and volume fractions (VEutectic) of eutectic cells that were dense in CHG: the tensile specimen E_2 with the lowest Voce parameters had the lower eutectic cell density, with VEutectic = 27.1%, whilst the tensile specimen C_2 with the highest Voce parameters had the higher density of eutectic cells, with VEutectic = 37.1%.
We are sorry to realize that the conclusions were not clearly underlined. So the number of pointed statements was reduced in order to make clear the achievements of this investigation in CONCLUSIONS, in page 25 of the MARKED_paper:
The findings of the study shed light on Defect Driven Plasticity (DDP), which refers to the unexpected regular plastic behavior of materials due to defects and metallurgical discontinuities, resulting in the linearity of Voce parameters in Matrix Assessment Diagrams (MADs) where the intercept of the best fitting line of Voce parameters is negative, which contradicts the physical meaning of the Voce equation parameters. Specifically, the study revealed qualitatively through fractography and quantitatively through mCT measurements that:
- The lowest positions of the Voce parameters in the linear data set correspond to the least defective microstructure, having also the longest elongations to rupture and highest UTS;
- The highest positions of the Voce parameters in the linear data set correspond to the most defective microstructure, also displaying the shortest elongations to rupture and smallest UTS.
A new additional finding was reported, resulting in a step forward into the comprehension of DDP:
- The linearity of the Voce parameters in MAD translates in the existence of a pivotal point belonging to all the experimental Voce differential curves, namely, (sP;QP);
- In sound materials, sP is higher than the saturation stresses sV of the tensile flow curves, and, as a consequence, the pivotal strain hardening rate QP is negative (non-real);
- In defective materials, the pivotal stress sP is between the mean YS and the saturation stresses sV, so that the strain hardening rate QP is positive (real);
- As material defectiveness increases, the mean yield strength (YS) and pivot stress (sP) become closer, causing the ratio of sP/YS to approach unity in defective castings; whilst with improving casting integrity, the ratio sP/YS increases well above the unity.
Reviewer’s comment 2. The author should try give more discussions about the current data and find some interesting phenomenon and mechanism.
Reply - In this paper we focused on chunky graphite. Indeed, a proper rationalization of DDP needs to gather also additional information on defective materials where other defects, like spiky graphite or porosities, for instance, could play a role. So other experimental efforts have to be faced to achieve hopefully some rationalization. However, some new important insights on DDP have been here reported.
Reviewer’ comment 3. In addition, there are too many SEM micrographs and they are not wll discussed.
Action – Thanks a lot for making us notice that our description of fracture surfaces was quite poor. We wrote additional description on fracture features (underlined in grey) at page 11 of the MARKED_paper:
3.3. Fractography
Figure 10(12) shows representative fracture surface micrographs of sample C_1 from the GJS400_P casting, taken through SEM with Secondary Electron Imaging (SEI). In Figure 10(12)a, a general view of the fracture surface reveals that the fracture was predominantly fibrous with signs of plastic deformation. Graphitic agglomerates in DIs operate as voids that later grow in dimensions with straining [43,44]; as the CHG size in the eutectic cells was very fine (see Figure 3), highly deformed cavities and small dimples were found, resulting in a noticeably fibrous fracture surface, as reported in Figure 10(12)b that provides details of a ductile fracture region at higher magnification. The density of CHG on the fracture surface appeared very high, consistently with the fact that the fracture path follow the graphitic agglomerates. The last fracture region, where cleavage facets fracture were visible indicating fracture by instability, is shown in Figure 10(12)c, and graphite nodules can be observed. The presence of very few shrinkage porosities on the fracture surface, as seen in Figure 10(12)d, suggests that CHG is probably the only significant defect in the material being investigated. Figure 11(13) shows representative fracture surface micrographs of sample C_2, also taken through SEM with SEI. In Figure 11(13)a, a general view of the fracture surface indicates that the fracture was generally ductile, which is also visible in Figure 11(13)b at higher magnification, where a high amount of CHG is present. The last fracture region is depicted in Figure 11(13)c, revealing cleavage facets and graphite nodules, and in Figure 11(13)d at higher magnification.

Reviewer 3 Report
-The term “Defects Driven Plasticity” needed to be describe considering heterogeneous structure of ductile iron (ferrite matrix, graphite, boundary). How defect driven plasticity differ from and defects driven fracture and were the microstructural boundary between these mechanisms. Please explain in introduction.
-Voce equations, and parameters mentioned in introduction needed to be presented there, not in research part, suggest made graphical illustration.
-Experimental part needs schematic of sample locations and marking
-Need better explanation how VFerrite, and VEutectic were determined from CT
-Need identify all symbols used in Eq.1
-From article: “In conclusion, we have confirmed the correlation between tensile properties, Voce parameters in MAD and IAD, and defective microstructure in castings”. Which particular graphs have claimed correlations with CT? There is no plotted Volumes from CT vs properties.
-Strain hardening depends on strain rate, this was not mentioned or discussed, suggest providing comments
-Conclusions sound controversial: The lowest positions of the Voce parameters in the linear data set correspond to the least defective microstructure, having also the shortest elongations to rupture and smallest UTS; The highest positions of the Voce parameters in the linear data set correspond to the most defective microstructure, also displaying the highest elongations to rupture." ???? Does this mean that more defect is better for material strength??? This part needed to be clarified because contradicts to common experience: less defects – better ductility and strength …
Author Response
Dear Reviewer,
thanks a lot for your comments and suggestions that have helped us to improve the quality of our paper.
Below you find the detailed reply to your comments and the actions we took to follow your suggestions. In the MARKED version of the paper you can read the text that concerns your suggestions underlined in yellow.
Kind regards
Giuliano Angella, PhD
………………………………………………..
Reviewer’s comment 1. The term “Defects Driven Plasticity” needed to be describe considering heterogeneous structure of ductile iron (ferrite matrix, graphite, boundary). How defect driven plasticity differ from and defects driven fracture and were the microstructural boundary between these mechanisms. Please explain in introduction.
Action – Thanks a lot for the interesting comment and useful suggestion. In Introduction at page we wrote to clarify what is expected in term effects of defects on plasticity …..
… neologism Defects Driven Plasticity (DDP). The regular plastic behavior of defective materials seems to suggest that a diffuse distribution of defects or metallurgical discontinuities, like CHG or porosity, for instance, should be responsible of this unexpected regular plastic behavior, according to the continuous damage mechanics where formation and growth of voids or cracks in the microstructure should occur during plastic deformation, forming a stable crack growth before the final rupture by instability [33,34]. Conversely, a single severe defect in a sound material could be also responsible of final rupture by instability because of stress intensification producing a brittle fracture [34]. In this second case the defect is expected to drive the fracture event itself rather than contributing to the whole plastic behavior. However, though the correlations between defective DIs and the intercepts of the best linear fits of Voce parameters has been established, so far the DDP has not been rationalized yet.
Reviewer’s comment 2. Voce equations, and parameters mentioned in introduction needed to be presented there, not in research part, suggest made graphical illustration.
Action – As suggested by the Reviewer we reported the plot of the Voce parameter determination in INTRODUCTION at page 3 of the MARKED_paper, so now the Voce parameters and fitting procedure are better described.
In a plot ds/deP vs. s, called Kocks and Mecking plot [3519,20,31,32] (here on called K-M plot), a data linearity can be found at high stresses well in the plastic regime corresponding to Stage III of strain hardening, and fitted with Equation 1b, so finding the constant Voce parameters Qo and 1/ec. Stage II of strain hardening is found only in single crystal experiments, whilst Stage II cannot be found in tensile tests, because deformation localization starts before the transition from Stage II to Stage IV [3532]. An example of Voce parameters calculation is reported in Figures 1(7) for a typical tensile flow curve of GJS400. In Figure 1a(7a), the procedure to find the Voce parameters in the K-M plot through the strain hardening analysis by using the differential Voce equation (Equation 1b) to fit the linear region of Stage III of strain hardening is reported, whilst in Figure 1b(7b) the corresponding tensile flow curve is shown. In the K-M plot, Q is the strain hardening rate (ds/deP), where s has the usual meaning.
Reviewer’s comment 3. Experimental part needs schematic of sample locations and marking.
Action – As suggested by the Reviewer in EXPERIMENTAL at page 6 of the MARKED_paper we reported a more schematic and clearer description of the tensile specimens extracted from the heavy castings:
The tensile specimens from the different castings were so taken:
- prismatic heavy section casting GJS400_P – two tensile specimens called C_1 and C_2 were machined off from the core of the block; two additional specimens, named S_1 and S_2, were taken from close to the external surface of the block, so materials from different cooling conditions could be tested;
- cylindrical heavy section casting GJS400_Ce – ten tensile specimens were machined off from a 25 mm thick slice at about 200 mm height of the cylindrical block with codes ranging from Nr 11 to Nr 20;
- 50 and 75 mm blocks GJS400_Y - details of the specimen selection and tensile testing are provided in [27,29,3031,36,37].
Reviewer’s comment 4. Need better explanation how VFerrite, and VEutectic were determined from CT.
Action – An extended and detailed description of the procedure and the algorithm used to find out the VFerrite and VEutectic has been reported in the text at page 5 of the MARKED_paper:
With mCT measurements it was possible to discriminate between metallic matrix and graphite agglomerates because of their different densities. However, due to the small size of CHG in eutectic cells located close to the instrumentation resolution and their high density in eutectic cells, it was difficult to quantify CHG agglomerates accurately. As a result, a different strategy was implemented. The eutectic phase full of fine CHG and the ferritic phase in the microtomographic image differed slightly in brightness: the former was darker, while the latter was brighter. The difficulty in identifying the phases was caused by the relatively high level of noise in the raw tomographic image. Therefore, the images were denoised using a median filter, and after denoising, a top-hat filter [41] was additionally applied, which highlighted the differences between the areas. Then, the images were thresholded to extract only the eutectic or only the ferritic part. The number of voxels in each part was then counted and multiplied by the voxel volume. The voxel volume is equal to the measurement resolution raised to the power of three. In this way, the volumes of each phase were determined. So the algorithm could discriminate between the two volumes with significantly different mean densities. The first volume, referred to as VFerrite, had a higher mean density and corresponded to the ferritic matrix with large embedded nodular graphitic agglomerates. The second volume, known as VEutectic, had a lower mean density and corresponded to the eutectic cells filled with CHG.
Reviewer’s comment 5. Need identify all symbols used in Eq.1
Action – Sorry, the variables were not explained; now they should be defined correctly at page 2 of the MARKED_paper underlined in green:
The analysis of the tensile strain hardening at room temperature in defective materials using the dislocation-density-related constitutive equation of Voce, which is based on physical principles, can provide valuable information regarding the defective nature of DIs [15,16,2719,20,31]. Voce constitutive equation is:
where s and eP are the true stress and plastic strain, respectively; sV is the saturation stress or maximum stress; eC is the critical plastic strain that defines the rate with which the saturation stress is achieved; and so is the back-extrapolated stress to zero plastic strain, close to yield stress.
Reviewer’s comment 6. From article: “In conclusion, we have confirmed the correlation between tensile properties, Voce parameters in MAD and IAD, and defective microstructure in castings”. Which particular graphs have claimed correlations with CT? There is no plotted Volumes from CT vs properties.
Action – As it was suggested correctly by the Reviewer, there were no summarizing plots of the achievements, so we introduced a figure (Figure 21 at page 21 of the MARKED_paper) and an explaining text of the figure:
In Figure 21a the fractography results and the Voce parameters in MAD from the GJS400_Ce casting are reported on the same plot. The fracture surface of the tensile specimen Nr 17 with the lowest Voce parameters presented a fracture surface with ferrite, nodular agglomerates and no CHG. The fracture surface of the tensile specimen Nr 20 with the highest Voce parameters shown a fracture surface completely covered in CHG (colored in brown for clarity purposes), whilst the tensile specimen Nr 16 with Voce parameter values in between, presented a fracture surface partially covered in CHG. In Figure 21b the mCT results and the Voce parameters in MAD from the GJS400_P casting shown the correlation between Voce parameters and volume fractions (VEutectic) of eutectic cells that were dense in CHG: the tensile specimen E_2 with the lowest Voce parameters had the lower eutectic cell density, with VEutectic = 27.1%, whilst the tensile specimen C_2 with the highest Voce parameters had the higher density of eutectic cells, with VEutectic = 37.1%.
Reviewer’s comment 7. Strain hardening depends on strain rate, this was not mentioned or discussed, suggest providing comments.
Reply – The Reviewer proposes an interesting issue. Strain hardening depends on strain rate, and DDP might be expected to depend too. Indeed, we have thought in the past that experiments on the effects of strain rates on defective materials could be interesting, but we have not planned them yet. Among the next possible investigations, strain rate could be an interesting parameter to be investigated, opening some new idea on the interpretation of DDP.
Reviewer’s comment 8. Conclusions sound controversial: The lowest positions of the Voce parameters in the linear data set correspond to the least defective microstructure, having also the shortest elongations to rupture and smallest UTS; The highest positions of the Voce parameters in the linear data set correspond to the most defective microstructure, also displaying the highest elongations to rupture." ???? Does this mean that more defect is better for material strength??? This part needed to be clarified because contradicts to common experience: less defects – better ductility and strength …
Action – It was a mistake, sorry. We corrected the text at page 25 of the MARKED_paper:
The findings of the study shed light on Defect Driven Plasticity (DDP), which refers to the unexpected regular plastic behavior of materials due to defects and metallurgical discontinuities, resulting in the linearity of Voce parameters in Matrix Assessment Diagrams (MADs) where the intercept of the best fitting line of Voce parameters is negative, which contradicts the physical meaning of the Voce equation parameters. Specifically, the study revealed qualitatively through fractography and quantitatively through mCT measurements that:
- The lowest positions of the Voce parameters in the linear data set correspond to the least defective microstructure, having also the shortest longest elongations to rupture and smallest highest UTS;
- The highest positions of the Voce parameters in the linear data set correspond to the most defective microstructure, also displaying the highest shortest elongations to rupture and smallest UTS.

Round 2
Reviewer 2 Report
The revised manuscript can be accepted as is.
Moderate editing of English language is needed
Reviewer 3 Report
No more comments